# Thymocytes trigger self-antigen-controlling pathways in immature medullary thymic epithelial stages

**Noella Lopes[1], Nicolas Boucherit[1†], Jérémy C Santamaria[1†], Nathan Provin[2†], Jonathan Charaix[1], Pierre Ferrier[1], Matthieu Giraud[2], Magali Irla[1*]**

[1]Aix-Marseille University, CNRS, INSERM, Centre d'Immunologie de Marseille-Luminy, Marseille, France; [2]Nantes Université, INSERM, Center for Research in Transplantation and Translational Immunology, UMR 1064, Nantes, France

**Abstract** Interactions of developing T cells with Aire[+] medullary thymic epithelial cells expressing high levels of MHCII molecules (mTEC[hi]) are critical for the induction of central tolerance in the thymus. In turn, thymocytes regulate the cellularity of Aire[+] mTEC[hi]. However, it remains unknown whether thymocytes control the precursors of Aire[+] mTEC[hi] that are contained in mTEC[lo] cells or other mTEC[lo] subsets that have recently been delineated by single-cell transcriptomic analyses. Here, using three distinct transgenic mouse models, in which antigen presentation between mTECs and CD4[+] thymocytes is perturbed, we show by high-throughput RNA-seq that self-reactive CD4[+] thymocytes induce key transcriptional regulators in mTEC[lo] and control the composition of mTEC[lo] subsets, including Aire[+] mTEC[hi] precursors, post-Aire and tuft-like mTECs. Furthermore, these inter-actions upregulate the expression of tissue-restricted self-antigens, cytokines, chemokines, and adhesion molecules important for T-cell development. This gene activation program induced in mTEC[lo] is combined with a global increase of the active H3K4me3 histone mark. Finally, we demon-strate that these self-reactive interactions between CD4[+] thymocytes and mTECs critically prevent multiorgan autoimmunity. Our genome-wide study thus reveals that self-reactive CD4[+] thymo-cytes control multiple unsuspected facets from immature stages of mTECs, which determines their heterogeneity.

## Editor's evaluation

This manuscript is of interest to readers in the field of immunology and especially in the induction of immune tolerance in the thymus. The work uses several mouse models to substantially broaden the current understanding of MHCII/TCR-mediated cell-cell crosstalk in the thymus and suggests a novel mechanism that contributes to the generation of functional and self-tolerant T-cells.

## Introduction

The thymic medulla ensures the generation of a self-tolerant T-cell repertoire (*Klein et al., 2014*; *Lopes et al., 2015*). By their unique ability to express tissue-restricted self-antigens (TRAs) (*Derbinski et al., 2001*; *Sansom et al., 2014*), medullary thymic epithelial cells (mTECs) promote the develop-ment of Foxp3[+] regulatory T cells and the deletion by apoptosis of self-reactive thymocytes capable of inducing autoimmunity (*Klein et al., 2019*). The expression of TRAs that mirrors body's self-antigens is controlled by Aire (*Autoimmune regulator*) and Fezf2 (*Fez family zinc finger 2*) transcription factors (*Anderson et al., 2002*; *Takaba et al., 2015*). Aire-dependent TRAs are generally characterized by a repressive chromatin state enriched in the trimethylation of lysine-27 of histone H3 (H3K27me3)

**\*For correspondence:**
Magali.Irla@inserm.fr

[†]These authors contributed equally to this work

**Competing interest:** The authors declare that no competing interests exist.

histone mark (*Handel et al., 2018*; *Org et al., 2009*; *Sansom et al., 2014*). In accordance with their essential role in regulating the expression of TRAs, *Aire*[-/-] and *Fezf2*[-/-] mice show defective clonal deletion of autoreactive thymocytes and develop signs of autoimmunity in several peripheral tissues (*Anderson et al., 2002*; *Takaba et al., 2015*).

Based on the level of the co-expressed MHC class II and CD80 molecules, mTECs were initially subdivided into mTEC[lo] (MHCII[lo]CD80[lo]) and mTEC[hi] (MHCII[hi]CD80[hi]) (*Gray et al., 2006*). The relationship between these two subsets has been established with reaggregate thymus organ cultures in which mTEC[lo] give rise to mature Aire[+] mTEC[hi] (*Gäbler et al., 2007*; *Gray et al., 2007*). Although mTEC[hi] express a highly diverse array of TRAs under Aire's action that releases stalled RNA polymerase and modulates chromatin accessibility, mTEC[lo] already express a substantial amount of TRAs (*Derbinski et al., 2005*; *Giraud et al., 2012*; *Koh et al., 2018*; *Kyewski and Klein, 2006*; *Sansom et al., 2014*). Recent single-cell transcriptomic analyses indicate that the heterogeneity of mTECs, especially in the mTEC[lo] compartment, is more complex than previously thought (*Irla, 2020*; *Kadouri et al., 2020*). mTEC[lo] with low or no expression of CD80 have been shown to be divided into three main subsets: CCL21[+] mTECs, implicated in the attraction of positively selected thymocytes in the medulla (*Lkhagvasuren et al., 2013*), involucrin[+]TPA[hi] post-Aire mTECs corresponding to the ultimate mTEC differentiation stage (*Metzger et al., 2013*; *Michel et al., 2017*; *Nishikawa et al., 2010*), and the newly reported tuft-like mTECs that show properties of gut chemosensory epithelial tuft cells expressing the doublecortin-like kinase 1 (DCLK1) marker (*Bornstein et al., 2018*; *Miller et al., 2018*). Based on single-cell transcriptomic analyses, mTECs were then classified into four major groups encompassing mTEC I:CCL21[+] mTECs, mTEC II:Aire[+] mTECs, mTEC III:post-Aire mTECs, and mTEC IV:tuft-like mTECs (*Bornstein et al., 2018*). Furthermore, mTEC[lo] with intermediate levels of CD80 and MHCII lie into mTEC single-cell clusters that are defined as proliferating and maturational, expressing Fezf2 and preceding the Aire[+] mTEC[hi] stage (*Baran-Gale et al., 2020*; *Dhalla et al., 2020*). These transit-amplifying cells were recently referred as to as TAC-TECs (*Wells et al., 2020*).

In the postnatal thymus, while mTECs control the selection of thymocytes, conversely CD4[+] thymocytes control the cellularity of Aire[+] mTEC[hi] by activating RANK and CD40-induced NF-κb signaling pathways (*Akiyama et al., 2008*; *Hikosaka et al., 2008*; *Irla, 2020*; *Irla et al., 2008*). These bidirectional interactions between mTECs and thymocytes are commonly referred to as thymic crosstalk (*Lopes et al., 2015*; *van Ewijk et al., 1994*). However, it remains unknown whether CD4[+] thymocytes act exclusively on mature Aire[+] mTEC[hi] or upstream on their TAC-TEC precursors contained in mTEC[lo] and whether the development of the newly identified Fezf2[+], post-Aire, and tuft-like subsets is regulated or not by CD4[+] thymocytes.

In this study, using high-throughput RNA-sequencing (RNA-seq), we show that self-reactive CD4[+] thymocytes induce in mTEC[lo] pivotal transcriptional regulators for their differentiation and function. Accordingly, self-reactive CD4[+] thymocytes control the composition of the mTEC[lo] compartment, that is the precursors of Aire[+] mTEC[hi], post-Aire cells, and tuft-like mTECs. Our data also reveal that self-reactive CD4[+] thymocytes upregulate in mTEC[lo] the expression of TRAs, chemokines, cytokines, and adhesion molecules involved in T-cell development. This gene activation program correlates with increased levels of the active trimethylation of lysine-4 of histone 3 (H3K4me3) mark, including the loci of Fezf2-dependent and Aire/Fezf2-independent TRAs, indicative of an epigenetic regulation for their expression. Finally, we demonstrate that disrupted MHCII/TCR interactions between mTECs and CD4[+] thymocytes lead to the generation of mature T cells containing self-specificities capable of inducing multiorgan autoimmunity. Altogether, our genome-wide study reveals that self-reactive CD4[+] thymocytes control the developmental transcriptional programs of mTEC[lo], which conditions their differentiation and function as inducers of T-cell tolerance.

## Results

### CD4[+] thymocytes induce key transcriptional programs in mTEC[lo] cells

Several NF-κb members are involved in Aire[+] mTEC[hi] development (*Burkly et al., 1995*; *Lomada et al., 2007*; *Riemann et al., 2017*; *Shen et al., 2019*; *Zhang et al., 2006*). However, it remains unclear whether the NF-κb or other signaling pathways are activated by CD4[+] thymocytes specifically in mTEC[lo] cells. To investigate the effects of CD4[+] thymocytes in mTEC[lo], we used mice deficient in CD4[+] thymocytes (ΔCD4 mice) because they lack the promoter IV of the class II transactivator

(*Ciita*) gene that controls MHCII expression in cortical TECs (cTECs) (*Waldburger et al., 2003*). We first analyzed by flow cytometry the total and phosphorylated forms of IKKα, p65, and RelB NF-κb members and p38 and Erk1/2 MAPK proteins in mTEC[lo] from ΔCD4 mice according to the gating strategy shown in *Figure 1—figure supplement 1A*. Interestingly, the phosphorylation level of IKKα and p38 MAPK was substantially reduced in ΔCD4 mice (*Figure 1A and B*, *Figure 1—figure supplement 2*), indicating that CD4[+] thymocytes may have an impact in mTEC[lo] by activating the IKKα intermediate of the nonclassical NF-κB pathway and the p38 MAPK pathway.

To gain insights into the effects of CD4[+] thymocytes in mTEC[lo], we analyzed by high-throughput RNA-seq the gene expression profiles of mTEC[lo] purified from WT and ΔCD4 mice (*Figure 1—figure supplement 1B*). We found that CD4[+] thymocytes upregulated 989 genes (fold change [FC] >2) reaching significance for 248 of them (Cuffdiff p<0.05) (*Figure 1C*). 957 genes were also downregulated (FC < 0.5) with 178 genes reaching significance (Cuffdiff p<0.05). We analyzed whether the genes significantly up- or downregulated by CD4[+] thymocytes corresponded to TRAs, as defined by an expression restricted to 1–5 of peripheral tissues (*Sansom et al., 2014*). Interestingly, the genes upregulated by CD4[+] thymocytes exhibited approximately fourfold more of TRAs over non-TRAs (*Figure 1D*, left panel). The comparison of the proportion of TRAs among the upregulated genes with those of the genome revealed a strong statistical TRA overrepresentation (p=5.2 × 10[-10]) (*Figure 1D*, right panel). Most of the TRAs upregulated by CD4[+] thymocytes were sensitive to the action of Aire (Aire-dependent TRAs) or controlled by Aire and Fezf2-independent mechanisms (Aire/Fezf2-independent TRAs) (*Figure 1E*, *Supplementary file 1*). The upregulation of some of these TRAs by CD4[+] thymocytes was confirmed by qPCR in mTEC[lo] purified from ΔCD4 mice (*Figure 1F*). The same results were observed with mTEC[lo] purified from MHCII[-/-] mice, also lacking CD4[+] thymocytes, excluding any potential indirect effect of CIITA in the phenotype observed in ΔCD4 mice (*Figure 1—figure supplement 3A*).

Remarkably, among the non-TRAs upregulated by CD4[+] thymocytes in mTEC[lo], 37 corresponded to 50 mTEC-specific transcription factors that are induced by the histone deacetylase 3 (HDAC3) (*Goldfarb et al., 2016*; *Figure 1G*). Some of them, such as the interferon regulatory factor 4 (*Irf4*), *Irf7*, and the Ets transcription factor member, *Spib,* are known to regulate mTEC differentiation and function (*Akiyama et al., 2014*; *Haljasorg et al., 2017*; *Otero et al., 2013*). We also identified other transcription factors such as *Nfkb2*, *Trp53,* and *Relb* implicated in mTEC differentiation (*Riemann et al., 2017*; *Rodrigues et al., 2017*; *Zhang et al., 2006*). Finally, we found that CD4[+] thymocytes upregulate in mTEC[lo] the expression of some cytokines and cell adhesion molecules such as integrins and cadherins (*Figure 1H*, *Figure 1—figure supplement 3B*). Given that mTEC[lo] are heterogeneous (*Irla, 2020*; *Kadouri et al., 2020*), we then analyzed whether the cytokines and adhesion molecules, which are upregulated by CD4[+] thymocytes, are specific to a particular subset of mTEC[lo]. To this end, we reanalyzed single-cell RNA-seq data performed on total CD45[-]EpCAM[+] TECs (*Wells et al., 2020*). Single cells were projected into a UMAP reduced-dimensional space and, using the 15 first principal components, six clusters were obtained, as in *Wells et al., 2020* (*Figure 1—figure supplement 4A*). Well-established markers were used to distinguish the different TEC subsets such as *Psmb11* and *Prss16* for cTECs, *Ccl21a* and *Krt5* for CCL21[+] mTECs (also called mTEC I), *Stmn1, Ska1, Fezf2* and *Aire* for TAC-TECs, *Aire* and *Fezf2* for Aire[+] mTECs (also called mTEC II), *Pigr* and *Cldn3* for post-Aire mTECs (also called mTEC III), and *Avil* and *Pou2f3* for tuft-like mTECs (also called mTEC IV) (*Figure 1—figure supplement 4B*). In contrast to CCL21[+] mTECs, some genes upregulated by CD4[+] thymocytes were expressed by tuft-like mTECs (*Figure 1I*). Interestingly, many genes encoding for cytokines and cell adhesion molecules were associated with Aire[+] mTECs and post-Aire cells with some of them already expressed in TAC-TECs, suggesting that CD4[+] thymocytes may act upstream of Aire[+] mTEC[hi]. These results thus provide the first evidence that CD4[+] thymocytes are able to induce in mTEC[lo] essential transcriptional regulators for mTEC differentiation and function as well as TRAs, adhesion molecules, and cytokines.

## CD4[+] thymocytes regulate maturational programs in mTEC[lo] through MHCII/TCR interactions

We next investigated by which mechanism CD4[+] thymocytes regulate the transcriptional programs of mTEC[lo]. Given that MHCII/TCR interactions with mTECs are critical for CD4[+] T-cell selection (*Klein et al., 2019*), we hypothesized that these interactions could play an important role in initiating

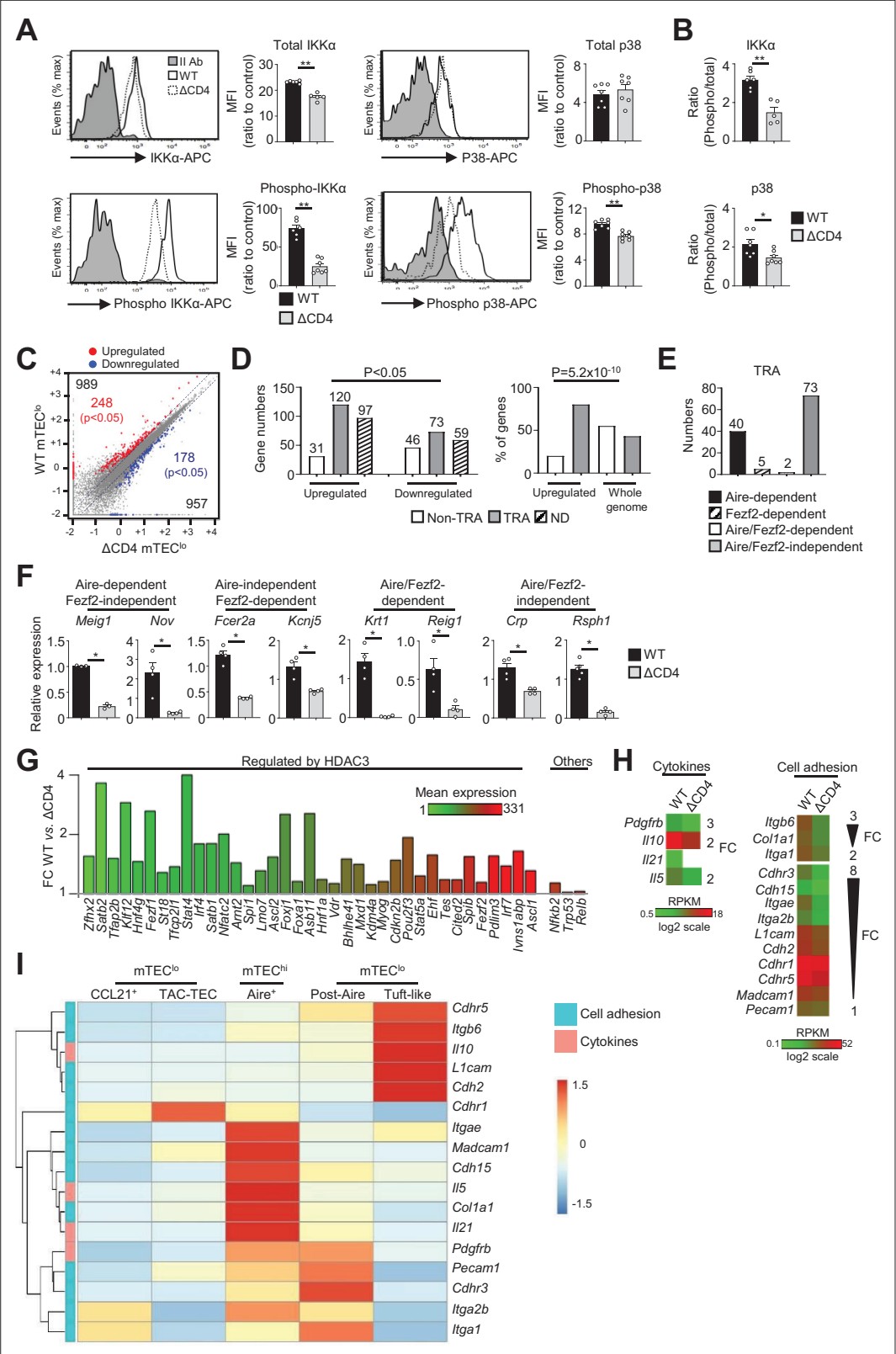

**Figure 1.** The transcriptional profile and IKKα and p38 MAPK signaling pathways are impaired in mTEC$^{lo}$ of ΔCD4 mice. (**A, B**) Total IKKα, p38 MAPK, phospho-IKKα(Ser180)/IKKβ(Ser181), and p38 MAPK (Thr180/Tyr182) (**A**) and the ratio of phospho/total proteins (**B**) analyzed by flow cytometry in mTEC$^{lo}$ from WT and ΔCD4 mice. Data are representative of two independent experiments (n = 3–4 mice per group and experiment). (**C**) Scatter

*Figure 1 continued on next page*

*Figure 1 continued*

plot of gene expression levels (fragments per kilobase of transcript per million mapped reads [FPKM]) of mTEC^lo from WT versus ΔCD4 mice. Genes with fold difference ≥2 and p-adj<0.05 were considered as upregulated or downregulated genes (red and blue dots, respectively). RNA-seq was performed on two independent biological replicates with mTEC^lo derived from 3 to 5 mice. (**D**) Numbers of tissue-restricted self-antigens (TRAs) and non-TRAs in genes up- and downregulated (left panel) and the proportion of upregulated TRAs compared to those in the all genome (right panel). ND, not determined. (**E**) Numbers of induced Aire-dependent, Fezf2-dependent, Aire/Fezf2-dependent, and Aire/Fezf2-independent TRAs. (**F**) The expression of Aire-dependent (*Meig1, Nov*), Fezf2-dependent (*Fcer2a, Kcnj5*), Aire/Fezf2-dependent (*Krt1, Reig1*), and Aire/Fezf2-independent (*Crp, Rsph1*) TRAs measured by qPCR in WT (n = 3–4) and ΔCD4 (n = 3–4) mTEC^lo. (**G**) Expression fold change in HDAC3-induced transcriptional regulators and other transcription factors significantly upregulated in WT versus ΔCD4 mTEC^lo. The color code represents gene expression level. (**H**) Heatmaps of genes encoding for cell adhesion molecules and cytokines that were significantly downregulated in mTEC^lo from ΔCD4 mice. (**I**) Hierarchical clustering and heatmap of mean expression of these cell adhesion molecules and cytokines in mTEC subsets identified by scRNA-seq. Error bars show mean ± SEM, *p<0.05, **p<0.01 using two-tailed Mann–Whitney test for (**A**), (**B**) and (**F**) and chi-squared test for (**D**).

The online version of this article includes the following figure supplement(s) for figure 1:

**Figure supplement 1.** Gating strategy used to purify mTEC^lo cells.

**Figure supplement 2.** Normal total and phosphorylated p65, RelB, and Erk1/2 proteins in mTEC^lo from ΔCD4 mice.

**Figure supplement 3.** Impaired TRA expression in mTEC^lo from MHCII^-/- mice.

**Figure supplement 4.** Identification of thymic epithelial cell (TEC) subsets by single-cell RNA-seq.

transcriptional programs that govern the functional and developmental properties of mTEC^lo. To this end, we used a unique transgenic mouse model in which MHCII expression is selectively abrogated in mTECs (mTEC^ΔMHCII mice) (*Irla et al., 2008*). In contrast to their WT counterparts, we found that OVA$_{323-339}$-loaded mTECs from mTEC^ΔMHCII mice were ineffective at activating OTII-specific CD4^+ T cells, demonstrating that the capacity of antigen presentation of mTECs to CD4^+ T cells is impaired in these mice (*Figure 2A*).

The comparison of the gene expression profiles of mTEC^lo purified from WT and mTEC^ΔMHCII mice (*Figure 1—figure supplement 1B*) revealed that MHCII/TCR interactions with CD4^+ thymocytes resulted in the upregulation of 1300 genes (FC > 2), 449 of them reaching statistical significance (Cuffdiff p<0.05). 846 genes were also downregulated (FC < 0.5) with 340 reaching significance (Cuffdiff p<0.05) (*Figure 2B*). Similarly to the comparison of WT versus ΔCD4 mice (*Figure 1D*), the genes significantly upregulated by MHCII/TCR interactions in mTEC^lo corresponded preferentially to TRAs (p=4.5 × 10^{-13}) that are mainly Aire-dependent and Aire/Fezf2-independent (*Figure 2C–E*, *Supplementary file 2*). In line with the recent discovery of *Aire* expression in mTECs expressing intermediate levels of CD80 identified in the proliferating and maturational stage mTEC single-cell clusters (*Dhalla et al., 2020*), we found a strong correlation (p=2 × 10^{-16}) between gene upregulation induced by MHCII/TCR interactions and the responsiveness of genes to Aire's action obtained from the comparison between WT and *Aire^-/-* mTEC^hi (*Figure 2F*). These data are in agreement with the identification of a list of activation factors including *Aire* among the non-TRA genes induced by MHCII/TCR interactions with CD4^+ thymocytes in mTEC^lo (*Figure 2G*). mTEC^lo from mTEC^ΔMHCII mice expressed ~4.5-fold less *Aire* than WT mTEC^lo, with substantial levels of 15.8 and 73.7 fragments per kilobase of transcript per million mapped reads (FPKM), respectively. For comparison, *Aire* expression level in WT mTEC^hi was 448.9 FPKM. mTEC^lo from mTEC^ΔMHCII mice also expressed ~1.5-fold less *Fezf2* than WT mTEC^lo (90.2 versus 134.5 FPKM, respectively). This reduction in *Aire* and *Fezf2* expression in mTEC^ΔMHCII mice was also confirmed by qPCR (*Figure 2H*). These results highlight the importance of MHCII/TCR interactions with CD4^+ thymocytes in upregulating *Aire* and *Fezf2* mRNAs and some of their associated TRAs in mTEC^lo. Interestingly, 17 HDAC3-regulated transcription factors as well as *Nfkb2*, *Trp53*, and *Relb* transcription factors were induced by MHCII/TCR interactions with CD4^+ thymocytes (*Figure 2I*). Moreover, the expression of several cytokines, chemokines, and cell adhesion molecules was also upregulated (*Figure 2J*, *Figure 2—figure supplement 1A*). Using single-cell RNA-seq data (*Figure 1—figure supplement 4*), we found that these genes were poorly associated with CCL21^+ and tuft-like mTEC^lo (*Figure 2K*). Consistently with the altered cellularity of Aire^+ mTECs in mTEC^ΔMHCII

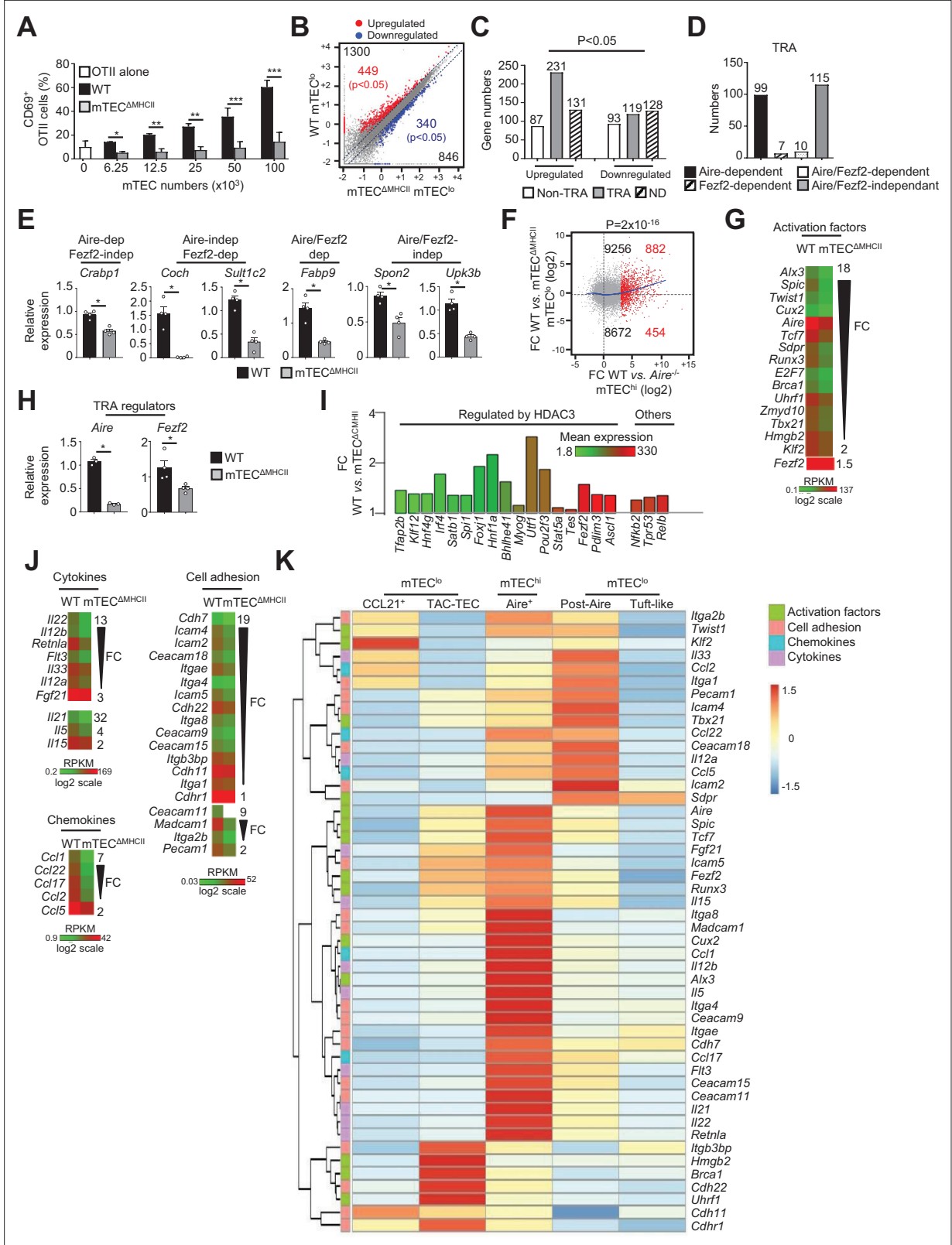

**Figure 2.** The transcriptional and functional properties of mTEC$^{lo}$ are impaired in mTEC$^{\Delta MHCII}$ mice. (**A**) Percentages of CD69$^+$ OTII CD4$^+$ T cells cultured or not with variable numbers of OVA$_{323-339}$-loaded WT or mTEC$^{\Delta MHCII}$ mTECs derived from two independent experiments (n = 2–3 mice per group and experiment). (**B**) Scatter plot of gene expression levels (fragments per kilobase of transcript per million mapped reads [FPKM]) of mTEC$^{lo}$ from WT versus mTEC$^{\Delta MHCII}$ mice. Genes with fold difference ≥2 and p-adj<0.05 were considered as upregulated or downregulated genes (red and blue dots,

*Figure 2 continued on next page*

Figure 2 continued

respectively). RNA-seq was performed on two independent biological replicates with mTEC$^{lo}$ derived from 3 to 5 mice. (C) Numbers of tissue-restricted self-antigens (TRAs) and non-TRAs in genes up- and downregulated in mTEC$^{lo}$ from WT versus mTEC$^{\Delta MHCII}$ mice. ND, not determined. (D) Numbers of induced TRAs regulated or not by Aire and/or Fezf2. (E) Aire-dependent (*Crabp1*), Fezf2-dependent (*Coch, Sult1c2*), Aire/Fezf2-dependent (*Fabp9*), and Aire/Fezf2-independent (*Spon2, Upk3b*) TRAs were measured by qPCR in mTEC$^{lo}$ from WT (n = 4) and mTEC$^{\Delta MHCII}$ (n = 4) mice. (F) Scatter plot of gene expression variation in mTEC$^{lo}$ from WT versus mTEC$^{\Delta MHCII}$ mice and in mTEC$^{hi}$ from WT versus *Aire*$^{-/-}$ mice. The loess fitted curve is shown in blue and the induced Aire-dependent genes (fold change [FC] > 5) in red. (G) Heatmap of significantly downregulated activation factors in mTEC$^{lo}$ from mTEC$^{\Delta MHCII}$ mice. (H) *Aire* and *Fezf2* mRNAs were measured by qPCR in mTEC$^{lo}$ from WT (n = 3–4) and mTEC$^{\Delta MHCII}$ (n = 4) mice. (I) FC in the expression of HDAC3-induced transcriptional regulators and other transcription factors significantly upregulated in WT versus mTEC$^{\Delta MHCII}$ mice. The color code represents gene expression level. (J) Heatmap of significantly downregulated cytokines, chemokines, and cell adhesion molecules in mTEC$^{lo}$ from mTEC$^{\Delta MHCII}$ mice. (K) Hierarchical clustering and heatmap of mean expression of these activation factors, cell adhesion molecules, chemokines, and cytokines in mTEC subsets identified by scRNA-seq. Error bars show mean ± SEM, *p<0.05, **p<0.01, ***p<0.001 using two-tailed Mann–Whitney test for (A), (E) and (H) and chi-squared test for (C) and (F).

The online version of this article includes the following figure supplement(s) for figure 2:

Figure supplement 1. Altered expression of some cytokines, cell adhesion molecules, and chemokines in mTEC$^{lo}$ from mTEC$^{\Delta MHCII}$ mice.

mice (*Irla et al., 2008*), some of these genes were associated with post-Aire cells. Strikingly, many genes upregulated by CD4$^+$ thymocytes in mTEC$^{lo}$ were highly expressed by Aire$^+$ mTECs. Interestingly, several of these genes were already expressed by TAC-TECs, including *Aire* and *Fezf2*, strongly suggesting that an enhanced transcriptional activity promoted by MHCII/TCR interactions with CD4$^+$ thymocytes accompanies the transition from TAC-TECs to Aire$^+$ mTECs. Altogether, these data show that CD4$^+$ thymocytes, through MHCII/TCR interactions, control the functional properties of mTEC$^{lo}$ and activate key transcriptional programs governing their differentiation and function.

## TCR/MHCII interactions with CD4$^+$ thymocytes regulate the development of Fezf2$^+$ pre-Aire, post-Aire, and tuft-like mTEC subsets

Since key transcription factors implicated in mTEC differentiation were upregulated in mTEC$^{lo}$ by MHCII/TCR-mediated interactions with CD4$^+$ thymocytes (*Figures 1G and 2I*), we next analyzed the composition for the newly identified mTEC subsets in ΔCD4 and mTEC$^{\Delta MHCII}$ mice. In agreement with our previous study (*Irla et al., 2008*), we first observed a substantial reduction in the frequencies and numbers of mTEC$^{hi}$ in both mice (*Figure 3A*). Furthermore, numbers of mTEC$^{lo}$ were also substantially reduced. Consequently, ΔCD4 and mTEC$^{\Delta MHCII}$ mice have a globally reduced cellularity in total mTECs. An Aire/Fezf2 co-staining both by histology and flow cytometry then revealed a substantial reduction in Aire$^-$Fezf2$^+$ and Aire$^+$Fezf2$^+$ cells (*Figure 3B and C*). We further analyzed by flow cytometry Aire and Fezf2 expression specifically in mTEC$^{lo}$ and mTEC$^{hi}$, according to the gating strategy shown in *Figure 1—figure supplement 1A*. In agreement with the detection of *Aire* in the proliferating and maturational single-cell clusters in mTEC$^{lo}$ (*Baran-Gale et al., 2020*; *Dhalla et al., 2020*; *Wells et al., 2020*), we found that Aire protein was expressed in a small fraction of mTEC$^{lo}$ compared to mTEC$^{hi}$ in WT, ΔCD4, and mTEC$^{\Delta MHCII}$ mice (*Figure 3C*). Aire$^-$Fezf2$^+$ and Aire$^+$Fezf2$^+$ mTECs were reduced in mTEC$^{lo}$ of ΔCD4 and mTEC$^{\Delta MHCII}$ mice with a more marked effect in mTEC$^{hi}$. This decrease was not due to impaired proliferation since normal frequencies of Ki-67$^+$ proliferating cells were observed in ΔCD4 and mTEC$^{\Delta MHCII}$ mice (*Figure 3—figure supplement 1*). Furthermore, numbers of involucrin$^+$TPA$^+$Aire$^-$ post-Aire cells were reduced in the medulla of ΔCD4 and mTEC$^{\Delta MHCII}$ mice (*Figure 3—figure supplement 2A*), consistently with the decrease of Aire$^+$ mTEC$^{hi}$ (*Figure 3B and C*). In contrast, the frequencies of CCL21$^+$ cells among mTEC$^{lo}$ were not altered in ΔCD4 and mTEC$^{\Delta MHCII}$ mice (*Figure 3D*). This is in line with the observation that few genes upregulated by TCR/MHCII interactions with CD4$^+$ thymocytes were associated with CCL21$^+$ mTECs (*Figures 1I and 2K*). We also analyzed tuft-like mTECs since the expression of the transcription factor *Pou2f3*, known to control the development of this cell type (*Bornstein et al., 2018*; *Miller et al., 2018*), was decreased in mTEC$^{lo}$ of ΔCD4 and mTEC$^{\Delta MHCII}$ mice (*Figures 1G and 2I*). We found that numbers of tuft-like mTECs identified by flow cytometry using the DCLK1 marker were reduced in both mice (*Figure 3E*, *Figure 3—figure supplement 2B*), indicating that their development is promoted by MHCII/TCR interactions with CD4$^+$ thymocytes. Importantly, Aire$^-$Fezf2$^+$ and Aire$^+$Fezf2$^+$ mTEC$^{lo}$ and mTEC$^{hi}$ as well as CCL21$^+$ and DCLK1$^+$ tuft-like mTEC$^{lo}$ were similarly reduced in MHCII$^{-/-}$ mice, further confirming that CD4$^+$ thymocytes control the cellularity of these novel mTEC subsets (*Figure 3—figure supplement 3*).

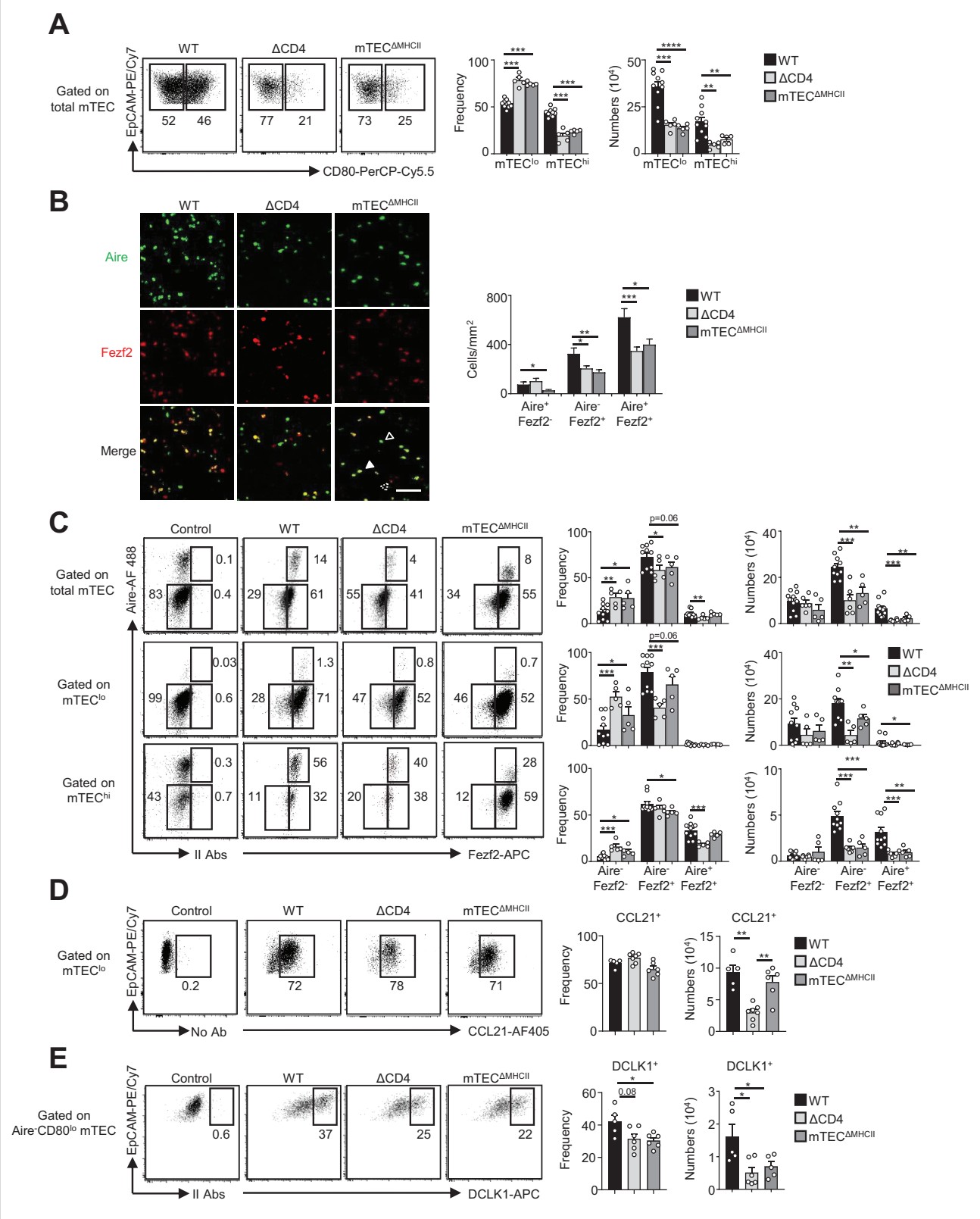

**Figure 3.** The composition in medullary thymic epithelial cell (mTEC) subsets is altered in ΔCD4 and mTEC^ΔMHCII mice. (**A**) Flow cytometry profiles, frequencies, and numbers of mTEC^lo and mTEC^hi in WT, ΔCD4, and mTEC^ΔMHCII mice. Data are representative of 2–3 independent experiments (n = 2–5 mice per group and experiment). (**B**) Confocal images of thymic sections from WT, ΔCD4, and mTEC^ΔMHCII mice stained for Aire (green) and Fezf2 (red). 12 and 20 sections derived from two WT, two ΔCD4, and two mTEC^ΔMHCII mice were quantified. Scale bar, 50 μm. Unfilled, dashed and

*Figure 3 continued on next page*

*Figure 3 continued*

solid arrowheads indicate Aire$^+$Fezf2$^-$, Aire$^-$Fezf2$^+$, and Aire$^+$Fezf2$^+$ cells, respectively. The histogram shows the density of Aire$^+$Fezf2$^-$, Aire$^-$Fezf2$^+$, and Aire$^+$Fezf2$^+$ cells. (**C–E**) Flow cytometry profiles, frequencies, and numbers of Aire$^-$Fezf2$^-$, Aire$^-$Fezf2$^+$, and Aire$^+$Fezf2$^+$ cells in total mTECs, mTEC$^{lo}$, and mTEC$^{hi}$ (**C**), of CCL21$^+$ cells in mTEC$^{lo}$ (**D**) and of DCKL1$^+$ cells in Aire$^-$ mTEC$^{lo}$ (**E**) from WT, ΔCD4, and mTEC$^{ΔMHCII}$ mice. II Abs: secondary antibodies. Data are representative of 2–3 independent experiments (n = 2–5 mice per group and experiment). Error bars show mean ± SEM, *p<0.05, **p<0.01, ***p<0.001, ****p<0.0001 using unpaired Student's *t*-test for (**B**) and two-tailed Mann–Whitney test for (**A**) and (**C**-**E**).

The online version of this article includes the following figure supplement(s) for figure 3:

**Figure supplement 1.** Normal proliferation of Aire$^-$Fezf2$^+$ and Aire $^+$Fezf2$^+$ medullary thymic epithelial cell (mTECs) in ΔCD4 and mTEC$^{ΔMHCII}$ mice.

**Figure supplement 2.** Reduced post-Aire medullary thymic epithelial cells (mTECs) in ΔCD4 and mTEC$^{ΔMHCII}$ mice.

**Figure supplement 3.** Analysis of medullary thymic epithelial cell (mTEC) subsets in MHCII$^{-/-}$ mice.

Altogether, these data reveal that MHCII/TCR-mediated interactions with CD4$^+$ thymocytes have a broad impact on mTEC composition by controlling the cellularity of not only Aire$^+$Fezf2$^+$ mTECs but also Fezf2$^+$ pre-Aire$^+$ mTECs, post-Aire, and tuft-like cells.

## Highly self-reactive CD4$^+$ thymocytes activate maturational programs in mTEC$^{lo}$

We next assessed the impact of highly self-reactive interactions with CD4$^+$ thymocytes in mTEC$^{lo}$ using OTII-*Rag2$^{-/-}$* and RipmOVAxOTII-*Rag2$^{-/-}$* transgenic mice. Both models possess CD4$^+$ thymocytes expressing an MHCII-restricted TCR specific for the chicken ovalbumin (OVA). The Rip-mOVA line expresses a membrane-bound OVA form specifically in mTECs, and consequently high-affinity interactions between OVA-expressing mTECs and OTII CD4$^+$ thymocytes are only possible in RipmOVAx-OTII-*Rag2$^{-/-}$* mice (*Kurts et al., 1996*). In contrast to total Erk1/2 MAPK, p38 MAPK, IKKα, and p65, the nonclassical NF-kB subunit RelB was increased in mTEC$^{lo}$ at mRNA and protein levels in RipmOVAx-OTII-*Rag2$^{-/-}$* compared to OTII-*Rag2$^{-/-}$* mice (*Figure 4A and B*, *Figure 4—figure supplement 1*). By reanalyzing single-cell RNA-seq data on mTEC$^{lo}$ subsets, we found that in contrast to CCL21$^+$ and tuft-like mTECs *Relb* is highly expressed by TAC-TECs and post-Aire cells, arguing again in favor that self-reactive CD4$^+$ thymocytes act from the TAC-TEC stage to induce their differentiation into Aire$^+$ cells and then into post-Aire cells (*Figure 4—figure supplement 2A*). The level of RelB phosphorylation was also higher in RipmOVAxOTII-*Rag2$^{-/-}$* than OTII-*Rag2$^{-/-}$* mice (*Figure 4B*), suggesting that self-reactive CD4$^+$ thymocytes may activate the nonclassical NF-κB pathway in mTEC$^{lo}$.

To define the genome-wide effects of highly self-reactive CD4$^+$ thymocytes in mTEC$^{lo}$, we compared the gene expression profiles of mTEC$^{lo}$ from RipmOVAxOTII-*Rag2$^{-/-}$* versus OTII-*Rag2$^{-/-}$* mice (*Figure 1—figure supplement 1B*) and found an upregulation of 1438 genes (FC > 2) reaching statistical significance for 522 of them (Cuffdiff p<0.05). 620 genes were also downregulated (FC < 0.5) with 136 reaching significance (Cuffdiff p<0.05) (*Figure 4C*). The genes upregulated exhibited an approximately fourfold more of TRA over non-TRA genes (p=4.7 × 10$^{-23}$), which corresponded mainly to Aire-dependent and Aire/Fezf2-independent TRAs (*Figure 4D–F*, *Supplementary file 3*). Similarly to the WT versus mTEC$^{ΔMHCII}$ comparison, we found a strong correlation (p=6.2 × 10$^{-7}$) between the genes upregulated by self-reactive CD4$^+$ thymocytes and the responsiveness of genes to Aire's action obtained from the comparison between WT and *Aire$^{-/-}$* mTEC$^{hi}$ (*Figure 4G*). These results support an impact of antigen-specific interactions in the expression of TRAs in mTEC$^{lo}$, notably on Aire-dependent TRAs. Importantly, these results are in agreement with the induction of a list of activation factors including *Aire* and *Fezf2* among the non-TRA genes (*Figure 4H*). Similarly to the comparisons of the WT versus ΔCD4 or mTEC$^{ΔMHCII}$ mice, numerous HDAC3-induced regulators as well as *Sirt1*, *Nfkb2*, *Relb*, and *Trp53* transcription factors were upregulated in mTEC$^{lo}$ of RipmOVAx-OTII-*Rag2$^{-/-}$* mice compared to OTII-*Rag2$^{-/-}$* mice (*Figure 4I*). Interestingly, 21 out of 30 top targets of the Foxn1 transcription factor, implicated in TEC differentiation and growth (*Žuklys et al., 2016*), as well as cytokines, chemokines, and cell adhesion molecules, were also upregulated (*Figure 4J*, K *Figure 4—figure supplement 2B*). We found that few of these genes were associated with CCL21$^+$ and tuft-like mTECs (*Figure 4L*). In contrast, many genes encoding for activation factors, cytokines, chemokines, and cell adhesion molecules were associated with Aire$^+$ and post-Aire mTECs, consistently with the fact that antigen-specific interactions with CD4$^+$ thymocytes control the cellularity of Aire$^+$ mTECs. Moreover, most of these genes, including *Aire* and *Fezf2*, were already expressed by

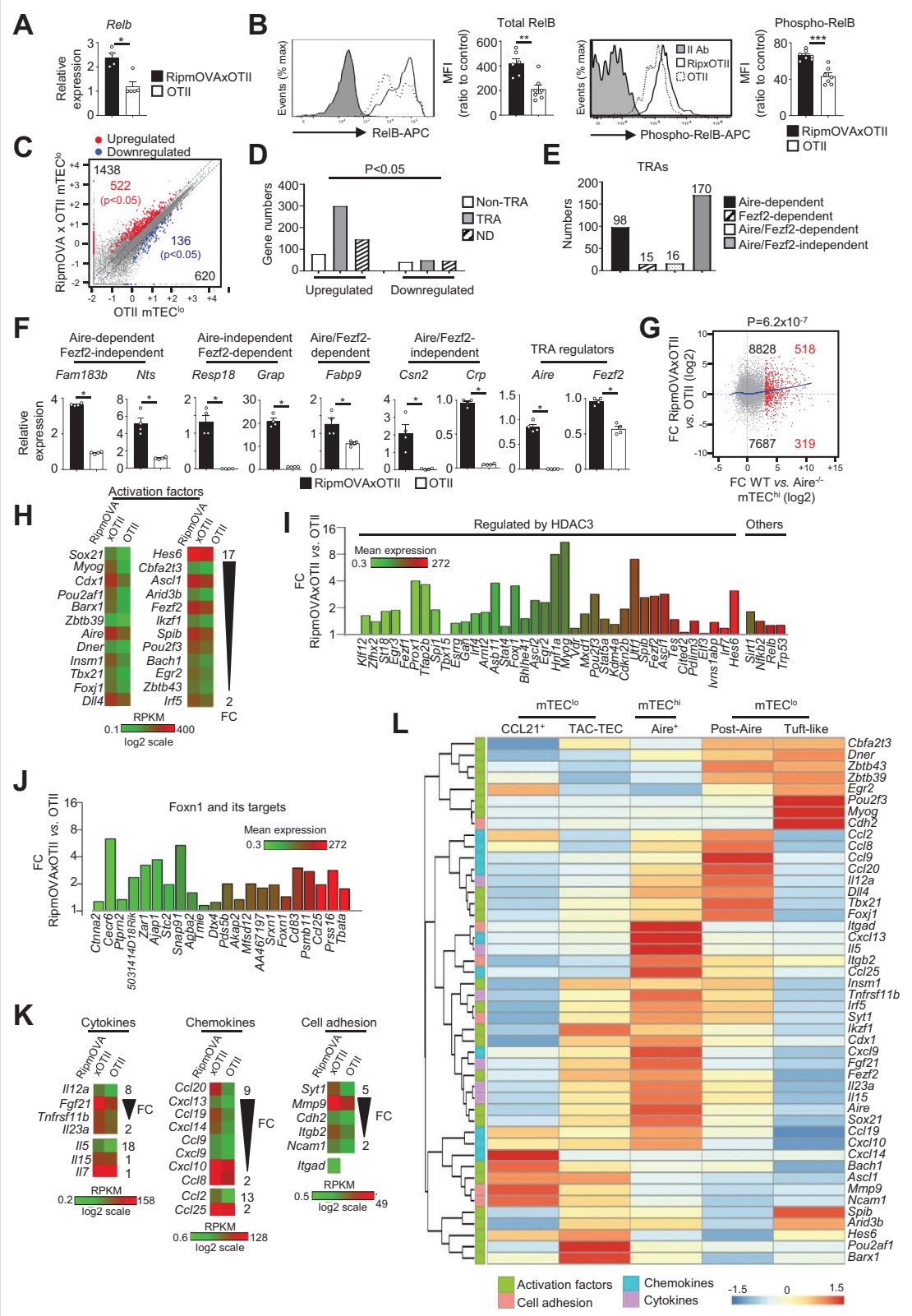

**Figure 4.** Highly self-reactive CD4[+] thymocytes control the transcriptional and functional properties of mTEC[lo]. (**A**) *Relb* mRNA was measured by qPCR in mTEC[lo] from RipmOVAxOTII-*Rag2*[-/-] (n = 4) and OTII-*Rag2*[-/-] (n = 5) mice. (**B**) Total and phospho-RelB (Ser552) were analyzed by flow cytometry in mTEC[lo] from RipmOVAxOTII-*Rag2*[-/-] and OTII-*Rag2*[-/-] mice. Data are representative of two independent experiments (n = 3–4 mice per group and experiment). (**C**) Scatter plot of gene expression levels (fragments per kilobase of transcript per million mapped reads [FPKM]) in mTEC[lo] from

*Figure 4 continued on next page*

*Figure 4 continued*

RipmOVAxOTII-*Rag2*[-/-] versus OTII-*Rag2*[-/-] mice. Genes with fold difference ≥2 and p-adj<0.05 were considered as upregulated or downregulated genes (red and blue dots, respectively). RNA-seq was performed on two independent biological replicates with mTEC[lo] derived from 5 to 8 mice. (**D**) Numbers of tissue-restricted self-antigens (TRAs) and non-TRAs in genes up- and downregulated in mTEC[lo] from RipmOVAxOTII-*Rag2*[-/-] versus OTII-*Rag2*[-/-] mice. ND, not determined. (**E**) Numbers of induced Aire-dependent, Fezf2-dependent, Aire/Fezf2-dependent, and Aire/Fezf2-independent TRAs. (**F**) Aire-dependent (*Fam183b, Nts*), Fezf2-dependent (*Resp18, Grap*), Aire/Fezf2-dependent (*Fabp9*), Aire/Fezf2-independent (*Csn2, Crp*) TRAs, *Aire* and *Fezf2* mRNAs were measured by qPCR in mTEC[lo] from RipmOVAxOTII-*Rag2*[-/-] (n = 4) and OTII-*Rag2*[-/-] (n = 4) mice. (**G**) Scatter plot of gene expression variation in mTEC[lo] from RipmOVAxOTII-*Rag2*[-/-] versus OTII-*Rag2*[-/-] mice and in mTEC[hi] from WT versus *Aire*[-/-] mice. The loess fitted curve is shown in blue and induced Aire-dependent genes (fold change [FC] > 5) in red. (**H**) Heatmap of significantly upregulated activation factors in mTEC[lo] from RipmOVAxOTII-*Rag2*[-/-] compared to OTII-*Rag2*[-/-] mice. (**I, J**) Expression FC in HDAC3-induced transcriptional regulators and other transcription factors (**I**) and in Foxn1 targets (**J**) in mTEC[lo] from RipmOVAxOTII-*Rag2*[-/-] versus OTII-*Rag2*[-/-] mice. The color code represents gene expression level. (**K**) Heatmap of significantly upregulated cytokines, chemokines, and cell adhesion molecules in mTEC[lo] from RipmOVAxOTII-*Rag2*[-/-] mice. (**L**) Hierarchical clustering and heatmap of mean expression of these activation factors, cell adhesion molecules, chemokines, and cytokines in mTEC subsets identified by scRNA-seq. Error bars show mean ± SEM, *p<0.05, **p<0.01, ***p<0.001 using two-tailed Mann–Whitney test for (**A**), (**B**) and (**F**) and chi-squared test for (**D**) and (**G**).

The online version of this article includes the following figure supplement(s) for figure 4:

**Figure supplement 1.** Similar levels of total and phosphorylated p65, Erk1/2, p38, and IKKα proteins in mTEC[lo] from RipmOVAxOTII-*Rag2*[-/-] and OTII-*Rag2*[-/-] mice.

**Figure supplement 2.** Expression of *Relb*, cytokines, chemokines, and cell adhesion molecules that was altered in mTEC[lo] from RipmOVAxOTII-*Rag2*[-/-] and OTII-*Rag2*[-/-] mice.

TAC-TECs, further highlighting that CD4[+] thymocytes act upstream of Aire[+] mTEC[hi]. Altogether, these data reveal that highly self-reactive CD4[+] thymocytes control in mTEC[lo] not only key transcription factors driven by their differentiation but also key molecules for T-cell development and selection such as TRAs, cytokines, chemokines, and adhesion molecules.

## Highly self-reactive CD4[+] thymocytes control mTEC subset composition from a progenitor stage

Given that highly self-reactive CD4[+] thymocytes induce key transcription factors in mTEC[lo] (***Figure 4H and I***), we examined their respective impact on mTEC subset development. Strikingly, numbers of total TECs and mTECs were higher in RipmOVAxOTII-*Rag2*[-/-] than in OTII-*Rag2*[-/-] mice (***Figure 5A and B***). We analyzed four TEC subsets based on MHCII and UEA-1 levels, as previously described (***Lopes et al., 2017***; ***Wong et al., 2014***; ***Figure 5C***). In contrast to cTEC[hi] (MHCII[hi]UEA-1[lo]), numbers of TEC[lo] (MHCII[lo]UEA-1[lo]), mTEC[lo] (MHCII[lo]UEA-1[+]), and mTEC[hi] (MHCII[hi]UEA-1[+]) were higher in RipmOVAx-OTII-*Rag2*[-/-] than in OTII-*Rag2*[-/-] mice. Consistently, numbers of mTEC[lo] and mTEC[hi] identified based on the level of CD80 expression were also higher in RipmOVAxOTII-*Rag2*[-/-] mice (***Figure 5—figure supplement 1***). Interestingly, numbers of α6-integrin[hi]Sca-1[hi] thymic epithelial progenitor (TEPC)-enriched cells in the TEC[lo] subset were also increased (***Figure 5D***), indicating that self-reactive CD4[+] thymocytes control TEC development from a progenitor stage. Of note, this strategy of TEC identification was not possible in ΔCD4 and mTEC[ΔMHCII] mice since MHCII expression is abrogated in TECs of these mice (***Irla et al., 2008***).

A higher density of Aire[+]Fezf2[-], Aire[-]Fezf2[+], and Aire[+]Fezf2[+] cells was observed in medullary regions of RipmOVAxOTII-*Rag2*[-/-] mice by immunohistochemistry (***Figure 5E***). Furthermore, numbers of Aire[-]Fezf2[-] mTEC[lo] analyzed by flow cytometry were also higher in RipmOVAx-OTII-*Rag2*[-/-] mice, confirming that self-reactive CD4[+] thymocytes control mTEC differentiation from an early stage (***Figure 5F***). Numbers of Aire[-]Fezf2[+] and Aire[+]Fezf2[+] mTEC[lo] and mTEC[hi] were also markedly increased in these mice, although similar frequencies of proliferating Ki-67[+] cells were observed (***Figure 5—figure supplement 2***). In agreement with increased Aire[+] mTECs, involucrin[+]T-PA[+]Aire[-] post-Aire cells were enhanced (***Figure 5—figure supplement 3***). Furthermore, numbers of CCL21[+] and DCLK1[+] tuft-like cells in mTEC[lo] were also increased in RipmOVAxOTII-*Rag2*[-/-] mice compared to OTII-*Rag2*[-/-] mice (***Figure 5G and H***). These observations are consistent with our previous findings that antigen-specific interactions between mTECs and CD4[+] thymocytes induce medulla development (***Irla et al., 2012***). Altogether, these results demonstrate that highly self-reactive CD4[+] thymocytes regulate mTECs from an early to a late developmental stage and thus mTEC composition.

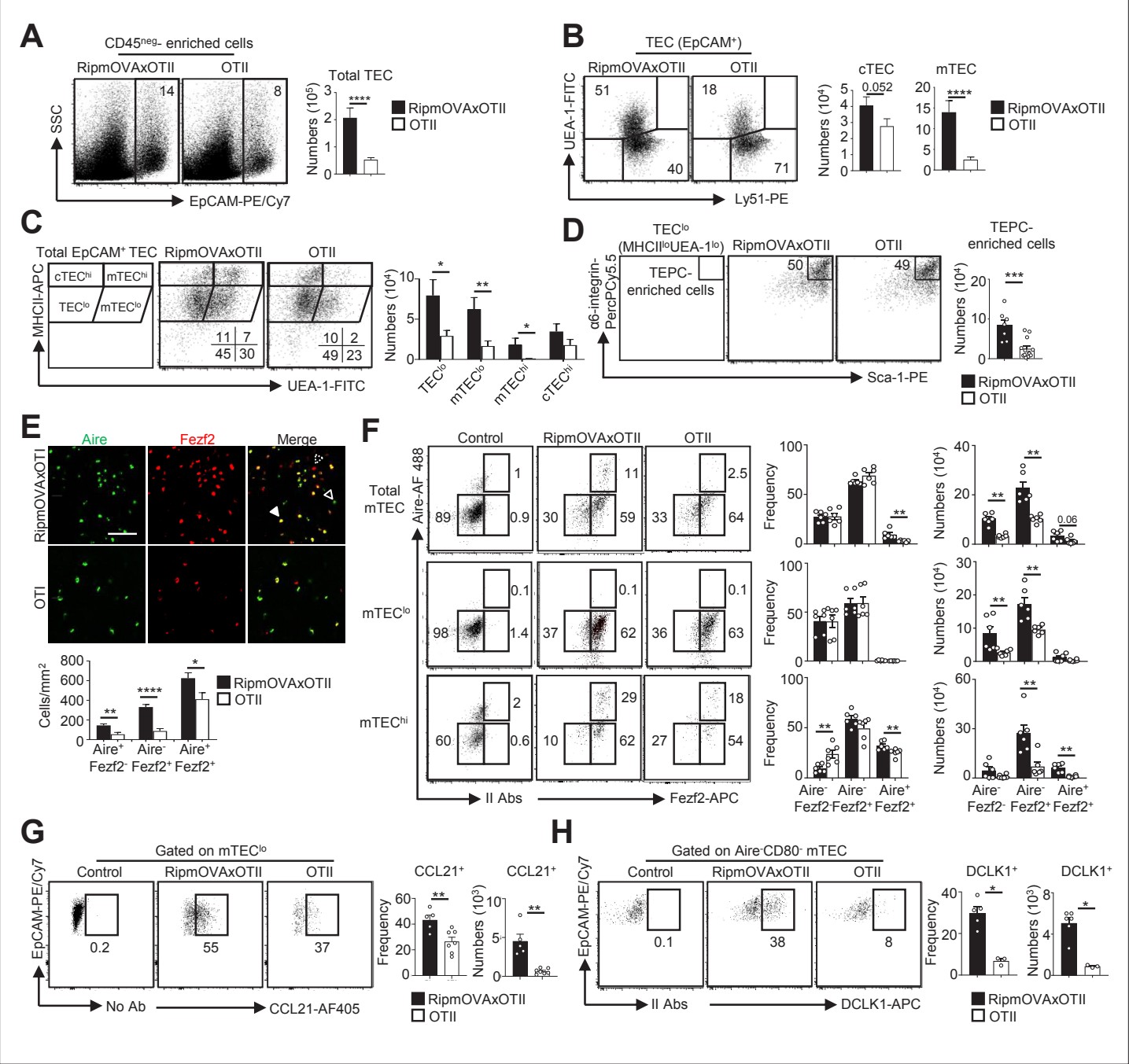

**Figure 5.** Highly self-reactive CD4+ thymocytes control medullary thymic epithelial cell (mTEC) development from an early progenitor stage. (**A–D**) Flow cytometry profiles and numbers of total thymic epithelial cells (TECs) (EpCAM+) (**A**), cortical thymic epithelial cell (cTECs) (UEA-1-Ly51hi), mTECs (UEA-1+Ly51lo) (**B**), TEClo (MHCIIloUEA-1lo), cTEChi (MHCIIhiUEA-1lo), mTEClo (MHCIIloUEA-1hi), and mTEChi (MHCIIhiUEA-1hi) (**C**), α6-integrinhiSca-1hi TEPC-enriched cells in TEClo (**D**) in CD45neg-enriched cells from RipmOVAxOTII-*Rag2-/-* and OTII-*Rag2-/-* mice. Data are representative of four experiments (n = 3 mice per group and experiment). (**E**) Confocal images of thymic sections from RipmOVAxOTII-*Rag2-/-* and OTII-*Rag2-/-* mice stained for Aire (green) and Fezf2 (red). 11 and 22 sections derived from two RipmOVAxOTII-*Rag2-/-* and OTII-*Rag2-/-* mice were quantified, respectively. Scale bar, 50 μm. Unfilled, dashed and solid arrowheads indicate Aire+Fezf2lo, Aire-Fezf2+, and Aire+Fezf2+ cells, respectively. The histogram shows the density of Aire+Fezf2lo, Aire-Fezf2+, and Aire+Fezf2+ cells. (**F–H**) Flow cytometry profiles, frequencies, and numbers of Aire-Fezf2-, Aire-Fezf2+, and Aire+Fezf2+ cells in total mTECs, mTEClo, and mTEChi (**F**), of CCL21+ cells in mTEClo (**G**) and of DCKL1+ cells in Aire- mTECs (**H**) from RipmOVAxOTII-*Rag2-/-* and OTII-*Rag2-/-* mice. II Abs: secondary antibodies. Data are representative of two independent experiments (n = 3–4 mice per group and experiment). Error bars show mean ± SEM, *p<0.05, **p<0.01, ***p<0.001, ****p<0.0001 using unpaired Student's *t*-test for (**A–E**) and two-tailed Mann–Whitney test for (**F–H**).

The online version of this article includes the following figure supplement(s) for figure 5:

*Figure 5 continued on next page*

*Figure 5 continued*

**Figure supplement 1.** mTEC$^{lo}$ and mTEC$^{hi}$ cells are increased in RipmOVAxOTII-*Rag2*$^{-/-}$ compared to OTII-*Rag2*$^{-/-}$ mice.

**Figure supplement 2.** The proliferation of Aire$^{-}$Fezf2$^{+}$ and Aire$^{+}$Fezf2$^{+}$ medullary thymic epithelial cells (mTECs) is similar in RipmOVAxOTII-*Rag2*$^{-/-}$ and OTII-*Rag2*$^{-/-}$ mice.

**Figure supplement 3.** Post-Aire medullary thymic epithelial cells (mTECs) are increased in RipmOVAxOTII-*Rag2*$^{-/-}$ compared to OTII-*Rag2*$^{-/-}$ mice.

## Self-reactive CD4$^{+}$ thymocytes enhance the level of active H3K4me3 mark in mTEC$^{lo}$

Since histone modifications constitute important regulatory mechanisms that control the open and closed states of mTEC chromatin (**Ucar and Rattay, 2015**), we investigated whether self-reactive CD4$^{+}$ thymocytes induce histone modifications in mTECs. We first analyzed in WT mTEC$^{lo}$ the repressive H3K27me3 and the active H3K4me3 marks using chromatin immunoprecipitation (ChIP) followed by high-throughput sequencing (ChIP-seq). As expected, metagene analyses showed that Aire-dependent TRAs had higher levels of H3K27me3 in their genes than in all genes of the genome, confirming that they are in a repressive state (**Figure 6A**). In contrast, Fezf2-dependent TRAs had a significant enrichment of H3K4me3 in their transcriptional start site (TSS) (**Figure 6B**). Similarly, Aire/Fezf2-independent TRAs were associated with low levels of H3K27me3 in their genes and high levels of H3K4me3 in their TSS. Thus, in contrast to Aire-dependent TRAs that are associated with the repressive H3K27me3 histone mark, Fezf2-dependent and Aire/Fezf2-independent TRAs are associated with the active H3K4me3 mark, indicating that these distinct TRAs are subjected to a specific epigenetic regulation.

We next assessed whether highly self-reactive CD4$^{+}$ thymocytes control the H3K27me3 and H3K4me3 chromatin landscape in mTEC$^{lo}$. In contrast to H3K27me3, we found an increased global level of H3K4me3 in RipmOVAxOTII-*Rag2*$^{-/-}$ compared to OTII-*Rag2*$^{-/-}$ mice by flow cytometry (**Figure 6C**). We further analyzed by nano-ChIP-seq whether self-reactive CD4$^{+}$ thymocytes regulate in mTEC$^{lo}$ the level of these two histone marks in Aire-dependent, Fezf2-dependent, and Aire/Fezf2-independent TRA genes. H3K27me3 levels in Aire-dependent TRA genes were comparable in mTEC$^{lo}$ from RipmOVAxOTII-*Rag2*$^{-/-}$ and OTII-*Rag2*$^{-/-}$ mice (**Figure 6D**, left panel). Although lower, H3K27me3 levels in Fezf2-dependent and Aire/Fezf2-independent TRAs as well as in all genes were similar in both mice, indicating that the interactions with self-reactive CD4$^{+}$ thymocytes do not regulate this repressive mark in TRA genes (**Figure 6D**, left panel). In contrast, H3K4me3 global level was increased in the TSS of all TRAs in RipmOVAxOTII-*Rag2*$^{-/-}$ compared to OTII-*Rag2*$^{-/-}$ mice as well as in all genes (**Figure 6D**, right panel). For representation, whereas the Aire/Fezf2-independent TRA, E2F transcription factor 2 (*E2f2*) induced by these interactions, was barely devoid of H3K27me3 in both mice, it was marked by H3K4me3 in its TSS specifically in RipmOVAxOTII-*Rag2*$^{-/-}$ mice (**Figure 6E**). These results thus show that self-reactive CD4$^{+}$ thymocytes enhance the global level of the active H3K4me3 histone mark in mTEC$^{lo}$ and in particular in the TSS of Fezf2-dependent and Aire/Fezf2-independent TRAs, indicative of an epigenetic regulation for their expression.

## MHCII/TCR interactions between mTECs and CD4$^{+}$ thymocytes prevent the development of autoimmunity

We next evaluated the impact of mTEC-CD4$^{+}$ thymocyte interactions on the generation of self-tolerant T cells by taking advantage that CD4$^{+}$ and CD8$^{+}$ T cells develop in mTEC$^{\Delta MHCII}$ mice, in which MHCII/TCR interactions between mTECs and CD4$^{+}$ thymocytes are abrogated. Interestingly, since TRAs induced by MHCII/TCR interactions showed a diverse peripheral tissue distribution in mTEC$^{lo}$ (**Figure 7A**, **Supplementary file 4**), we analyzed the TCRVβ usage in mTEC$^{\Delta MHCII}$ mice by flow cytometry. TCRVβ usage was more altered in CD69$^{-}$ mature CD4$^{+}$ thymocytes than in CD8$^{+}$ thymocytes (**Figure 7B**). Some TCRVβ were also altered in splenic CD4$^{+}$ and CD8$^{+}$ T cells. To determine whether these T cells contained self-reactive specificities, we adoptively transferred splenocytes from mTEC$^{\Delta MHCII}$ or WT mice into lymphopenic *Rag2*$^{-/-}$ recipients (**Figure 7C**). Mice that received splenocytes derived from mTEC$^{\Delta MHCII}$ mice lost significantly more weight than mice transferred with WT splenocytes (**Figure 7D**). They also exhibited splenomegaly with increased follicle areas and T-cell numbers showing a CD62L$^{lo}$CD44$^{hi}$ effector and CD62L$^{hi}$CD44$^{hi}$ central memory phenotype (**Figure 7E–G**). Immune infiltrates in lungs and salivary glands were observed by histology and flow cytometry in 75

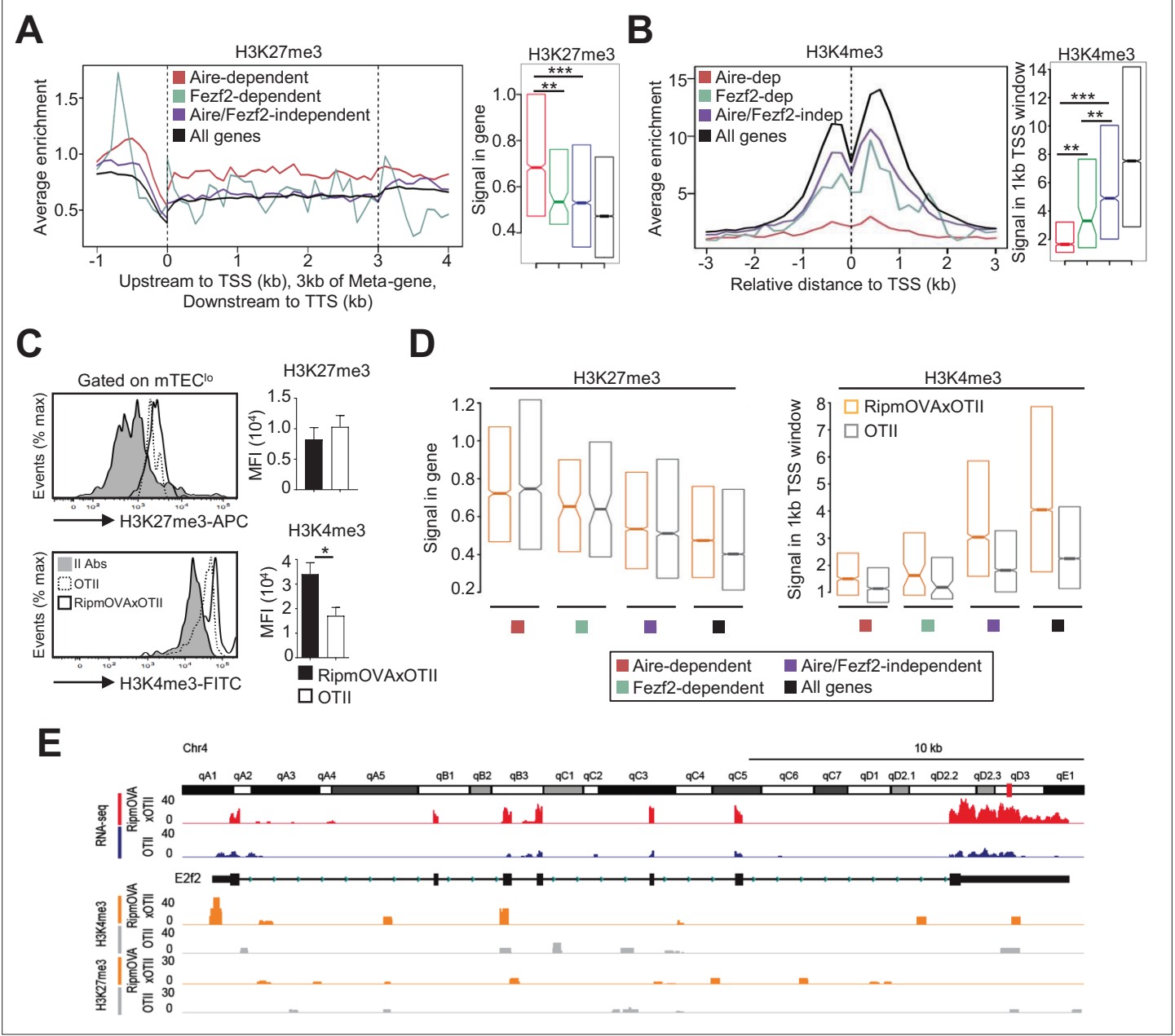

**Figure 6.** H3K27me3 and H3K4me3 landscape in tissue-restricted self-antigen (TRA) genes of mTEC$^{lo}$ from WT, RipmOVAxOTII-*Rag2*$^{-/-}$, and OTII-*Rag2*$^{-/-}$ mice. (**A, B**) Metagene profiles of the average normalized enrichment of H3K27me3 (**A**) and H3K4me3 (**B**) against input for Aire-dependent, Fezf2-dependent, and Aire/Fezf2-independent TRAs as well as for all genes of WT mTEC$^{lo}$. Boxplots represent the median enrichment, the 95% CI of the median (notches), and the 75th and 25th percentiles of H3K27me3 and H3K4me3. (**C**) H3K27me3 and H3K4me3 levels were analyzed by flow cytometry in mTEC$^{lo}$ from RipmOVAxOTII-*Rag2*$^{-/-}$ (n = 4) and OTII-*Rag2*$^{-/-}$ (n = 5) mice. Histograms show the MFI. (**D**) Boxplots represent the median enrichment of H3K27me3 and H3K4me3 of Aire-dependent, Fezf2-dependent, and Aire/Fezf2-independent TRAs and in all genes of mTEC$^{lo}$ from RipmOVAxOTII-*Rag2*$^{-/-}$ and OTII-*Rag2*$^{-/-}$ mice. (**E**) Expression (RNA-seq) and H3K27me3 and H3K4me3 chromatin state (ChIP-seq) of the Aire/Fezf2-independent TRA, *E2f2*, in mTEC$^{lo}$ from RipmOVAxOTII-*Rag2*$^{-/-}$ and OTII-*Rag2*$^{-/-}$ mice. *p<0.05, **p<10$^{-3}$, ***p<10$^{-7}$, using the Mann–Whitney test for (**A**) and (**B**) and unpaired Student's *t*-test for (**C**).

and 41% of mice, respectively (***Figure 7H and I***). These two tissues contained increased numbers of central memory as well as CD44$^+$CD69$^+$ and CD44$^+$CD69$^-$ activated CD4$^+$ and CD8$^+$ T cells (***Figure 7J***). T-cell infiltrates were also observed in other tissues such as kidney, liver, and colon in agreement with the defective TRA expression associated with these tissues (***Figure 7A and K***). Altogether, these data show that in the absence of MHCII/TCR interactions between mTECs and CD4$^+$ thymocytes, T cells

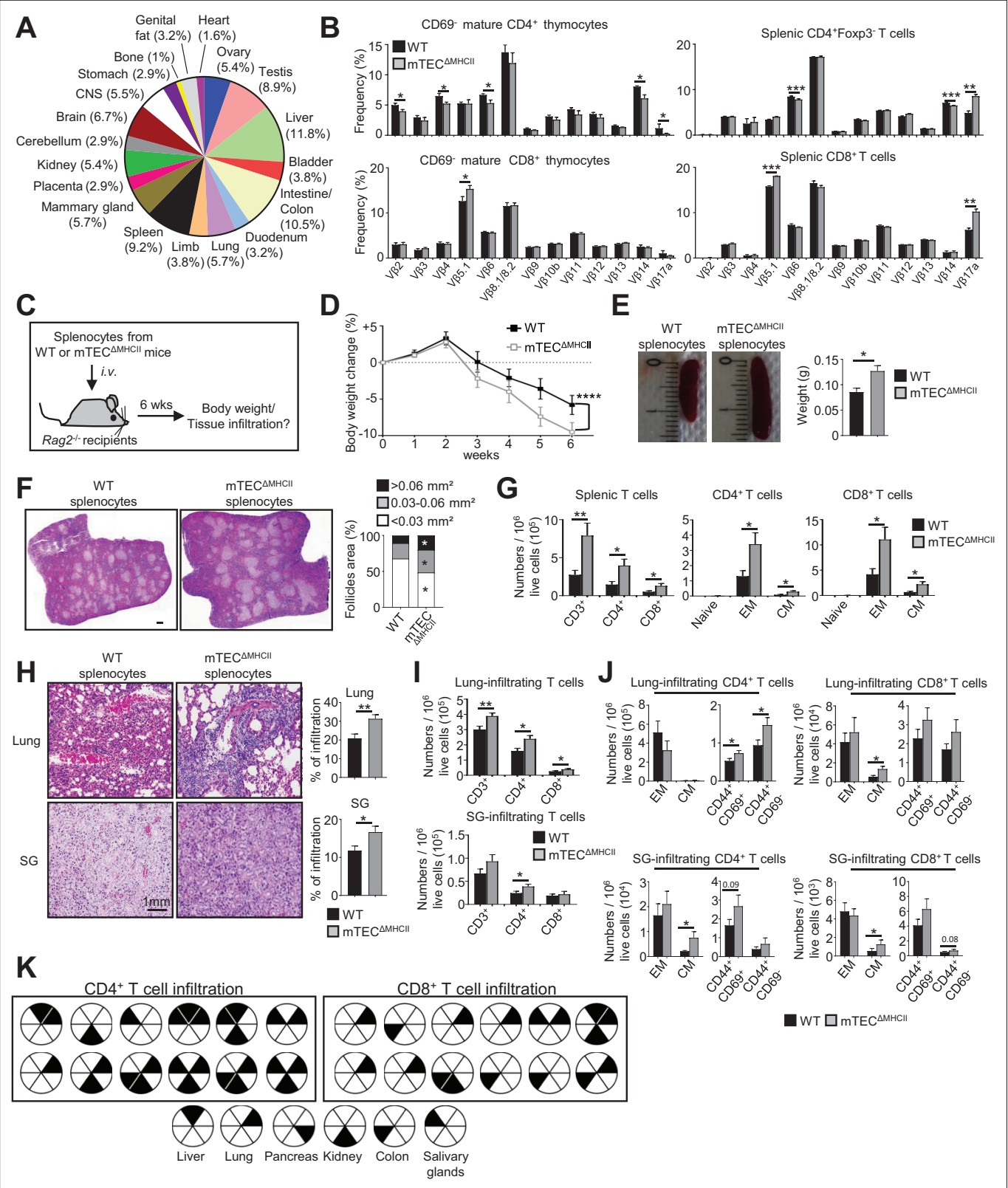

**Figure 7.** The adoptive transfer of T cells from mTEC^ΔMHCII mice into *Rag2^-/-* recipients induces autoimmunity. (**A**) Tissue-restricted self-antigens (TRAs) underexpressed in mTEC^lo from mTEC^ΔMHCII mice were assigned to their peripheral expression. (**B**) TCRVβ usage by CD69^- mature CD4^+ and CD8^+ thymocytes (left panel) and CD4^+Foxp3^- and CD8^+ splenic T cells (right panel) from WT and mTEC^ΔMHCII mice. (**C**) Body weight of *Rag2^-/-* recipients transferred with splenic T cells from WT or mTEC^ΔMHCII was monitored during 6 weeks, and tissue infiltration was examined. (**D**) Weight loss relative

*Figure 7 continued on next page*

*Figure 7 continued*

to the initial weight. (**E, F**) Representative spleen pictures and their weights (**E**) and hematoxylin/eosin counterstained splenic sections (**F**). Scale bar, 1 mm. The histogram shows follicle areas. (**G**) Numbers of splenic CD3$^+$, CD4$^+$, and CD8$^+$ T cells and of naive (CD44$^{lo}$CD62L$^{hi}$), effector memory (EM; CD44$^{hi}$CD62L$^{lo}$) and central memory (CM; CD44$^{hi}$CD62L$^{hi}$) phenotype. (**H**) Lung and salivary gland (SG) immune infiltrates detected by hematoxylin/eosin counterstaining. Scale bar, 1 mm. (**I, J**) Numbers of T cells (**I**) and of naive, effector and central memory phenotype as well as CD44$^+$CD69$^+$ and CD44$^+$CD69$^-$ T cells (**J**) in lungs and SG. (**K**) Schematic of T-cell infiltrates in mice transferred with mTEC$^{\Delta MHCII}$ T cells relative to those transferred with WT T cells. Each circle and black triangles represent an individual mouse and T-cell infiltration in a specific tissue, respectively. Data are representative of two independent experiments (n = 5–7 mice per group and experiment). Error bars show mean ± SEM, ****p<0.0001 using two-way ANOVA for (**D**) and unpaired Student's *t*-test for (**B**) and (**E**-**J**). *p<0.05, **p<0.01, ***p<0.001.

The online version of this article includes the following figure supplement(s) for figure 7:

**Figure supplement 1.** CD4$^+$ thymocytes through MHCII/TCR-mediated interactions control transcriptional programs of mTEC$^{lo}$ that drive their differentiation and function.

contained self-reactive specificities and thus that these interactions are critical to the establishment of T-cell tolerance.

## Discussion

Since mTECs play a crucial role in immunological tolerance by their exclusive expression of TRAs, it is essential to deepen our knowledge of the mechanisms that sustain their differentiation. Here using three distinct transgenic models, we found that self-reactive CD4$^+$ thymocytes control the developmental transcriptional programs from the mTEC$^{lo}$ stage, including TAC-TECs that precede Aire$^+$ mTECs. CD4$^+$ thymocytes increase in mTEC$^{lo}$ the phosphorylation of p38 MAPK and IKKα, the latter implicated in mTEC development (*Lomada et al., 2007*; *Shen et al., 2019*). Moreover, self-reactive CD4$^+$ thymocytes increase RelB phosphorylation level. Interestingly, this nonclassical NF-κB subunit is crucial for mTEC differentiation and Aire-dependent and -independent TRA expression (*Riemann et al., 2017*). These data thus suggest that CD4$^+$ thymocytes activate intracellular pathways from the mTEC$^{lo}$ stage, although alterations in mTEC$^{lo}$ subset composition could also contribute to the differences observed. Nevertheless, the substantial and homogeneous reduction in the levels of phospho-IKKα, -p38, and -RelB argues instead for impaired activation of IKKα, p38, and RelB signaling in the absence of self-reactive CD4$^+$ thymocytes. Analysis of the mTEC$^{lo}$ transcriptional landscape by high-throughput RNA-seq revealed that self-reactive CD4$^+$ thymocytes upregulate *Nfkb2* (p52), known to form an heterocomplex with RelB in the nucleus upon activation (*Irla et al., 2010*). p52 is important for mTEC development, Aire, and TRA expression (*Zhang et al., 2006*; *Zhu et al., 2006*). Consequently, ΔCD4, mTEC$^{\Delta MHCII}$, and OTII-*Rag2$^{-/-}$* mice in which MHCII/TCR interactions between mTECs and CD4$^+$ thymocytes are disrupted have altered *Relb* and *Nfkb2* expression, and reduced Aire$^+$ mTEC numbers and Aire-dependent TRA representation. Our results are in agreement with the fact that RANK-induced NF-κB signaling is activated by membrane-bound RANKL and not soluble RANKL and thereby in the context of physical interactions between mTECs and CD4$^+$ thymocytes (*Asano et al., 2019*).

These interactions also upregulate *Trp53* (p53) that controls the mTEC niche (*Rodrigues et al., 2017*) and *Irf4* and *Irf7* transcription factors that regulate key chemokines implicated in thymocyte medullary localization and mTEC differentiation (*Haljasorg et al., 2017*; *Otero et al., 2013*). Furthermore, the deacetylase Sirtuin-1 (*Sirt1*), which regulates Aire activity (*Chuprin et al., 2015*), and *Spib*, which limits mTEC differentiation (*Akiyama et al., 2014*), were also upregulated. Self-reactive CD4$^+$ thymocytes thus induce key transcription factors that both positively and negatively control mTEC differentiation. Remarkably, our three different transgenic models revealed that CD4$^+$ thymocytes induce HDAC3-dependent mTEC-specific transcription factors (*Goldfarb et al., 2016*). Among them, *Pou2f3* is involved in tuft-like mTEC development (*Bornstein et al., 2018*; *Miller et al., 2018*), which is consistent with our results showing that self-reactive CD4$^+$ thymocytes control the cellularity of these cells. Our data thus identify that CD4$^+$ thymocytes control the expression of master transcriptional regulators of mTEC differentiation and function.

In line with these data, we found that self-reactive CD4$^+$ thymocytes regulate TEC development from a progenitor stage since they increase numbers of TEPC-enriched cells that express non-negligible MHCII levels. Interestingly, we provide the first evidence that self-reactive CD4$^+$ thymocytes

control the cellularity of Fezf2$^+$ mTECs. Accordingly, the expression of Fezf2 and its respective TRAs was enhanced by CD4$^+$ thymocytes. Moreover, self-reactive CD4$^+$ thymocytes regulate the cellularity of CCL21$^+$, post-Aire, and tuft-like cells in mTEC$^{lo}$. These results are in full agreement with our previous findings that self-reactive thymocytes drive medulla expansion and increase the overall cellularity of the mTEC compartment (*Irla et al., 2012*). Because the heterogeneous composition in mTEC$^{lo}$ could influence the expression of the upregulated genes by self-reactive CD4$^+$ thymocytes, we reanalyzed single-cell RNA-seq data in order to define their respective expression pattern in mTEC subsets. In accordance with the moderately altered frequencies of CCL21$^+$ cells among mTEC$^{lo}$ observed by flow cytometry, few genes upregulated by self-reactive CD4$^+$ thymocytes were associated with this mTEC subset. In contrast, we found that antigen-specific interactions with CD4$^+$ thymocytes strongly upregulate genes associated with TAC-TECs, Aire$^+$ mTECs, and post-Aire cells. These findings indicate that self-reactive CD4$^+$ thymocytes act from the precursors of Aire$^+$ mTEC$^{hi}$ (i.e., in TAC-TECs) to the post-Aire stage. It is interesting to note that although strongly altered the development of mTEC$^{hi}$ is not completely abrogated in the absence of CD4$^+$ thymocytes or MHCII/TCR-mediated interactions with CD4$^+$ thymocytes. This could be explained by the fact that invariant NKT have been proposed to participate in mTEC differentiation by expressing RANKL (*White et al., 2014*). Overall, our results thus reveal that antigen-specific interactions with CD4$^+$ thymocytes have an unsuspected broad impact on mTEC composition by driving their development from an early progenitor to a late post-Aire stage.

Interestingly, high-throughput RNA-seq showed that MHCII/TCR interactions with CD4$^+$ thymocytes upregulate the expression of chemokines in mTEC$^{lo}$. Among them, CCL19 (CCR7 ligand) is implicated in the medullary localization of thymocytes and the emigration of newly generated T cells (*Ueno et al., 2004*); and CCL22 (CCR4 ligand) implicated in medullary entry and thymocyte/dendritic cell interactions (*Hu et al., 2015*). Self-reactive CD4$^+$ thymocytes also enhance CCL2 (CCR2 ligand) and CCL20 (CCR6 ligand) that promote the entry of peripheral dendritic cells and Foxp3$^+$ regulatory T cells into the thymus (*Baba et al., 2009*; *Cédile et al., 2014*; *Cowan et al., 2018*; *Lopes et al., 2018*; *Borelli and Irla, 2021*). mTEC-CD4$^+$ thymocyte interactions thus induce key chemokines that regulate the trafficking of thymocytes and dendritic cells that participate in tolerance induction. Moreover, cytokines such as *Il15* and *Fgf21* implicated in invariant NKT development and TEC protection against senescence, as well as adhesion molecules involved in mTEC-thymocyte interactions, were also induced (*Pezzi et al., 2016*; *White et al., 2014*; *Youm et al., 2016*). Altogether, our data show that self-reactive CD4$^+$ thymocytes regulate functional properties of mTECs by inducing chemokines, cytokines, and adhesion molecules that are critical for T-cell development.

The expression of TRAs is regulated by Aire and to a lesser extent by Fezf2 (*Anderson et al., 2002*; *Takaba et al., 2015*). In agreement with other studies (*Gray et al., 2007*; *Takaba et al., 2015*), we found Fezf2 in both mTEC$^{lo}$ and mTEC$^{hi}$, whereas Aire protein is mainly expressed in mTEC$^{hi}$. Nevertheless and in line with recent single-cell transcriptomic analyses (*Baran-Gale et al., 2020*; *Dhalla et al., 2020*; *Wells et al., 2020*), we detected Aire by flow cytometry, qPCR, and RNA-seq in a small subset (~1.5%) of mTEC$^{lo}$. CD4$^+$ thymocyte interactions upregulate *Aire* and *Fezf2* and some of their respective TRAs in these cells. Interestingly, in contrast to Aire-dependent TRAs that are characterized by high levels of H3K27me3 (*Handel et al., 2018*; *Sansom et al., 2014*), we found that Fezf2-dependent TRAs show high levels of H3K4me3. This highlights that Aire and Fezf2 use distinct epigenetic modes in regulating TRA expression. Remarkably, these interactions also induce in mTEC$^{lo}$ numerous Aire/Fezf2-independent TRAs, whose regulation remains unknown. Similarly to Fezf2-dependent TRAs, they had high levels of H3K4me3 in their TSS, suggesting that Aire/Fezf2-independent TRAs are not subjected to the same regulatory transcriptional mechanisms than Aire-dependent TRAs. Our results are consistent with a previous study indicating that the Aire-independent TRA, *Gad1*, shows active epigenetic marks (*Tykocinski et al., 2010*). Remarkably, self-reactive CD4$^+$ thymocytes increase H3K4me3 level in the TSS of all TRA categories, thus providing a novel epigenetic mechanistic insight into how they regulate the mTEC gene expression profile. In line with TRA regulation and the development of distinct mTEC subsets, the repertoire of mature T cells contains autoreactive cells when MHCII/TCR interactions were abrogated between mTECs and CD4$^+$ thymocytes. Accordingly, the adoptive transfer of splenocytes from mTEC$^{\Delta MHCII}$ mice is capable of inducing signs of autoimmunity, illustrating the fact that mTEC-CD4$^+$ thymocyte interactions are critical for the generation of a self-tolerant T-cell repertoire. Future investigations based on TCR sequencing analysis are expected to define to which extent the TCR repertoire is altered in mTEC$^{\Delta MHCII}$ mice.

In summary, our genome-wide scale study reveals that self-reactive CD4$^+$ thymocytes activate transcriptional programs from the TAC-TEC stage that sustains the differentiation into Aire$^+$Fezf2$^+$ and post-Aire mTECs (*Figure 7—figure supplement 1*). These interactions also upregulate the expression of TRAs, cytokines, chemokines, and adhesion molecules that are all implicated in mTEC function. Thus, CD4$^+$ thymocytes control several unsuspected aspects of mTEC$^{lo}$ required for the establishment of T-cell tolerance.

# Materials and methods

## Key resources table

| Reagent type (species) or resource | Designation | Source or reference | Identifiers | Additional information |
|---|---|---|---|---|
| Genetic reagent (*Mus musculus*) | C57BL/6J background | Charles River | RRID:IMSR_JAX:000664 | |
| Genetic reagent (*M. musculus*) | *Ciita*$^{tm2Wrth}$/*Ciita*$^{tm2Wrth}$ | *LeibundGut-Landmann et al., 2004* | RRID:MGI:3052466 | C57BL/6 background, ΔCD4 mice |
| Genetic reagent (*M. musculus*) | H2$^{dlAb1-Ea}$/H2$^{dlAb1-Ea}$ | *Madsen et al., 1999* | RRID:MGI:4436873 | C57BL/6 background, MHCII$^{-/-}$ mice |
| Genetic reagent (*M. musculus*) | K14x*Ciita*$^{III+IV-/-}$ | *Irla et al., 2008* | | C57BL/6 background, mTEC$^{ΔMHCII}$ mice |
| Genetic reagent (*M. musculus*) | Tg(TcraTcrb)425Cbn | *Barnden et al., 1998* | RRID:MGI:3762632 | C57BL/6 background, OTII mice |
| Genetic reagent (*M. musculus*) | Tg(Ins2-TFRC/OVA)296Wehi | *Kurts et al., 1996* | RRID:MGI:3623748 | C57BL/6 background, Rip-mOVA mice |
| Genetic reagent (*M. musculus*) | *Rag2*$^{tm1Fwa}$/*Rag2*$^{tm1Fwa}$ | *Shinkai et al., 1992* | RRID:MGI:2174910 | C57BL/6 background, *Rag2*$^{-/-}$ mice |
| Antibody | Anti-IKKα (rabbit polyclonal) | Cell Signaling Technology | Cat# 2682; RRID:AB_331626 | FACS (1:500) |
| Antibody | Anti-phospho IKKα (Ser180)/IKKβ(Ser181) (rabbit polyclonal) | Cell Signaling Technology | Cat# 2681S; RRID:AB_331624 | FACS (1:500) |
| Antibody | Anti-p38 MAPK (rabbit polyclonal) | Cell Signaling Technology | Cat# 9212; RRID:AB_330713 | FACS (1:500) |
| Antibody | Anti-phospho p38 MAPK (Thr180/Tyr182) (rabbit polyclonal) | Cell Signaling Technology | Cat# 9211S; RRID:AB_331641 | FACS (1:500) |
| Antibody | Anti-Erk1/2 (rabbit polyclonal) | Cell Signaling Technology | Cat# 9102; RRID:AB_330744 | FACS (1:500) |
| Antibody | Anti-phospho Erk1/2 (Thr202/Tyr204) (rabbit polyclonal) | Cell Signaling Technology | Cat# 9101S; RRID:AB_331646 | FACS (1:500) |
| Antibody | Anti-NF-$\kappa$B p65 (clone D14E12, rabbit monoclonal) | Cell Signaling Technology | Cat# 8242S; RRID:AB_10859369 | FACS (1:500) |
| Antibody | Phospho-NF-$\kappa$B p65 (Ser536) (clone 93H1, rabbit monoclonal) | Cell Signaling Technology | Cat# 3033S; RRID:AB_331284 | FACS (1:3000) |
| Antibody | Anti-RelB (clone C-19, rabbit polyclonal) | Santa Cruz Biotechnology | Cat# sc-226; RRID:AB_632341 | FACS (1:200) |
| Antibody | Anti-phospho RelB (ser552) (clone D41B9, rabbit monoclonal) | Cell Signaling Technology | Cat# 5025S; RRID:AB_10622001 | FACS (1:1000) |
| Antibody | Anti-DCLK1 (clone D2U3L, rabbit monoclonal) | Cell Signaling Technology | Cat# 62257; RRID:AB_2799622 | FACS (1:200) |
| Antibody | Anti-H3K4me3 (rabbit polyclonal) | Abcam | Cat# ab8580; RRID:AB_306649 | FACS (1:1000) ChIP-seq (2 µg:25 µg chromatin) |
| Antibody | Anti-H3K27me3 (clone C36B11, rabbit monoclonal) | Cell Signaling Technology | Cat# 9733; RRID:AB_2616029 | FACS (1:1000) ChIP-seq (1:50) |

*Continued on next page*

*Continued*

| Reagent type (species) or resource | Designation | Source or reference | Identifiers | Additional information |
|---|---|---|---|---|
| Antibody | PE-Cy7 anti-CD326 (EpCAM) (clone G8.8, rat monoclonal) | eBioscience | Cat# 25-5791-80; RRID:AB_1724047 | FACS (1:3000) |
| Antibody | Alexa Fluor 488 anti-Aire (clone 5H12, rat monoclonal) | eBioscience | Cat# 53-5934-82; RRID:AB_10854132 | FACS, IF (1:200) |
| Antibody | PE anti-Ly51 (clone BP-1, mouse monoclonal) | BD Biosciences | Cat# 553735; RRID:AB_395018 | FACS (1:3000) |
| Antibody | PerCP-Cy5.5 anti-CD80 (clone 16-10A1, Armenian hamster monoclonal) | BioLegend | Cat# 104722; RRID:AB_2291392 | FACS (1:200) |
| Antibody | eFluor 450 anti-Ki-67 (clone SolA15, rat monoclonal) | eBioscience | Cat# 48-5698-82; RRID:AB_11149124 | FACS (1:200) |
| Antibody | Anti-Fezf2 (clone F441, rabbit polyclonal) | IBL Tecan | Cat# JP18997; RRID:AB_2341444 | FACS, IF (1:200) |
| Antibody | Anti-Involucrin (clone Poly19244, rabbit polyclonal) | BioLegend | Cat# 924401; RRID:AB_2565452 | IF (1:100) |
| Antibody | PE anti-Ly-6A/E (Sca-1) (clone D7, rat monoclonal) | BD Biosciences | Cat# 553108; RRID:AB_394629 | FACS (1:600) |
| Antibody | Biotin anti-CD49f (α6-integrin) (clone GoH3, rat monoclonal) | BioLegend | Cat# 313604; RRID:AB_345298 | FACS (1:200) |
| Antibody | Alexa Fluor 647 anti-I-Ab (MHCII) (clone AF6-120.1, mouse monoclonal) | BioLegend | Cat# 116412; RRID:AB_493141 | FACS (1:200) |
| Antibody | Brilliant Violet 421 anti-CD4 (clone RM4-5, rat monoclonal) | BioLegend | Cat# 100544; RRID:AB_11219790 | FACS (1:200) |
| Antibody | PerCP-Cy5.5 anti-CD4 (clone RM4-5, rat monoclonal) | BD Biosciences | Cat# 550954; RRID:AB_393977 | FACS (1:200) |
| Antibody | Pacific Blue anti-CD8α (clone 53-6.7, rat monoclonal) | BD Biosciences | Cat# 558106; RRID:AB_397029 | FACS (1:200) |
| Antibody | PE/Cy7 anti-CD8α (clone 53-6.7, rat monoclonal) | BioLegend | Cat# 100722; RRID:AB_312761 | FACS (1:600) |
| Antibody | Alexa Fluor 488 anti-CD44 (clone IM7, rat monoclonal) | BioLegend | Cat# 103016; RRID:AB_493679 | FACS (1:200) |
| Antibody | PE anti-CD69 (clone H1.2F3, rat monoclonal) | BioLegend | Cat# 104508; RRID:AB_313111 | FACS (1:400) |
| Antibody | PE anti-CD62L (clone MEL-14, rat monoclonal) | BD Biosciences | Cat# 553151; RRID:AB_394666 | FACS (1:300) |
| Antibody | PerCP-Cy5.5 anti-CD3ε (clone 17A2, rat monoclonal) | BD Biosciences | Cat# 560527; RRID:AB_1727463 | FACS (1:200) |
| Antibody | Alexa Fluor 405 anti-CCL21 (clone 59106, rat monoclonal) | R&D Systems | Cat# IC457V | FACS (1:100) |
| Antibody | CD45 MicroBeads, mouse (clone 30F11.1, rat monoclonal) | Miltenyi | Cat# 130052301; RRID:AB_2877061 | |
| Antibody | Cy5 anti-rabbit IgG (goat polyclonal) | Invitrogen | Cat# A10523; RRID:AB_2534032 | FACS (1:500) |
| Antibody | Cyanine 3 anti-rabbit IgG (goat polyclonal) | Invitrogen | Cat# A10520; RRID:AB_2534029 | IF (1:500) |
| Antibody | FITC anti-TCR Vβ2 (clone B20.6, rat monoclonal) | BD Biosciences | Cat# 557004; RRID:AB_647180 | FACS (20 μl per 10⁶ cells) |
| Antibody | FITC anti-TCR Vβ3 (clone KJ25, Armenian hamster monoclonal) | BD Biosciences | Cat# 557004; RRID:AB_647180 | FACS (20 μl per 10⁶ cells) |

*Continued*

| Reagent type (species) or resource | Designation | Source or reference | Identifiers | Additional information |
|---|---|---|---|---|
| Antibody | FITC anti-TCR Vβ4 (clone KT4, rat monoclonal) | BD Biosciences | Cat# 557004; RRID:AB_647180 | FACS (20 µl per $10^6$ cells) |
| Antibody | FITC anti-TCR Vβ5.1, 5.2 (clone MR9-4, mouse monoclonal) | BD Biosciences | Cat# 553189; RRID:AB_394697 | FACS (1:100) |
| Antibody | FITC anti-TCR Vβ6 (clone RR4-7, rat monoclonal) | BD Biosciences | Cat# 557004; RRID:AB_647180 | FACS (20 µl per $10^6$ cells) |
| Antibody | PE anti-TCR Vβ8.1, 8.2 (clone MR5-2, mouse monoclonal) | BioLegend | Cat# 140103; RRID:AB_10641144 | FACS (1:300) |
| Antibody | FITC anti-TCR Vβ9 (clone MR10-2, mouse monoclonal) | BD Biosciences | Cat# 557004; RRID:AB_647180 | FACS (20 µl per $10^6$ cells) |
| Antibody | PE anti-TCR Vβ10b (clone B21.5, rat monoclonal) | BD Biosciences | Cat# 553285; RRID:AB_394757 | FACS (1:300) |
| Antibody | Biotin anti-TCR Vβ11 (clone RR3-15, rat monoclonal) | BD Biosciences | Cat# 553196; RRID:AB_394702 | FACS (1:300) |
| Antibody | FITC anti-TCR Vβ12 (clone MR11-1, mouse monoclonal) | BD Biosciences | Cat# 557004; RRID:AB_647180 | FACS (20 µl per $10^6$ cells) |
| Antibody | FITC anti-TCR Vβ13 (clone MR12-3, mouse monoclonal) | BD Biosciences | Cat# 557004; RRID:AB_647180 | FACS (20 µl per $10^6$ cells) |
| Antibody | FITC anti-TCR Vβ14 (clone 14-2, rat monoclonal) | BD Biosciences | Cat# 557004; RRID:AB_647180 | FACS (20 µl per $10^6$ cells) |
| Antibody | FITC anti-TCR Vβ17a (clone KJ23, rat monoclonal) | BD Biosciences | Cat# 557004; RRID:AB_647180 | FACS (20 µl per $10^6$ cells) |
| Antibody | CD4$^+$ T cell isolation kit, mouse | Miltenyi Biotec | Cat# 130-104-454 | |
| Peptide, recombinant protein | PerCP-Cy5.5 Streptavidin | BioLegend | Cat# 405214; RRID:AB_2716577 | FACS (1:400) |
| Peptide, recombinant protein | Alexa Fluor 488 Streptavidin | Invitrogen | Cat# S11223 | IF (1:1000) |
| Peptide, recombinant protein | Ovalbumin (323–339) | PolyPeptide | Cat# SC1303 | 5 µM |
| Chemical compound, drug | Liberase TM | Roche | Cat# 05401127001 | 50 µg/ml |
| Chemical compound, drug | DNase I | Roche | Cat# 10104159001 | 100 µg/ml |
| Chemical compound, drug | TRIzol | Thermo Fisher Scientific | Cat# 15596018 | |
| Software, algorithm | GraphPad Prism | GraphPad Software | RRID:SCR_002798 | |
| Software, algorithm | FlowJo | FlowJo | https://www.flowjo.com/ RRID:SCR_008520 | |
| Software, algorithm | Fiji/ImageJ software | Fiji-ImageJ | https://imagej.nih.gov/ij/ RRID:SCR_003070 | |
| Software, algorithm | 7500 Real-Time PCR Software | Thermo Fisher | https://www.thermofisher.com/us/en/home/technical-resources/software-downloads/applied-biosystems-7500-real-time-pcr-system.html RRID:SCR_014596 | |
| Software, algorithm | Pheatmap 0.2 | https://github.com/raivokolde/pheatmap (*Kolde, 2018*) | RRID:SCR_016418 | |
| Software, algorithm | Seurat | *Hao et al., 2021* | RRID:SCR_016341 | |

*Continued on next page*

*Continued*

| Reagent type (species) or resource | Designation | Source or reference | Identifiers | Additional information |
|---|---|---|---|---|
| Sequence-based reagent | Actin-FW | Sigma-Aldrich | PCR primers | CAGAAGGAGATTACTGCTCTGGCT |
| Sequence-based reagent | Actin-RV | Sigma-Aldrich | PCR primers | GGAGCCACCGATCCACACA |
| Sequence-based reagent | Aire-FW | Sigma-Aldrich | PCR primers | GCATAGCATCCTGGACGGCTTCC |
| Sequence-based reagent | Aire-RV | Sigma-Aldrich | PCR primers | CTGGGCTGGAGACGCTCTTTGAG |
| Sequence-based reagent | Ccl19-FW | Sigma-Aldrich | PCR primers | GCTAATGATGCGGAAGACTG |
| Sequence-based reagent | Ccl19-RV | Sigma-Aldrich | PCR primers | ACTCACATCGACTCTCTAGG |
| Sequence-based reagent | Ccl2-FW | Sigma-Aldrich | PCR primers | TGGAGCATCCACGTGTTG |
| Sequence-based reagent | Ccl2-RV | Sigma-Aldrich | PCR primers | ACTCATTGGGATCATCTTGCT |
| Sequence-based reagent | Ccl22-FW | Sigma-Aldrich | PCR primers | CTGATGCAGGTCCCTATGGT |
| Sequence-based reagent | Ccl22-RV | Sigma-Aldrich | PCR primers | GGAGTAGCTTCTTCACCCAG |
| Sequence-based reagent | Ccl25-FW | Sigma-Aldrich | PCR primers | GCCTGGTTGCCTGTTTTGTT |
| Sequence-based reagent | Ccl25-RV | Sigma-Aldrich | PCR primers | ACCCAGGCAGCAGTCTTCAA |
| Sequence-based reagent | Cdh2-FW | Sigma-Aldrich | PCR primers | AGCGCAGTCTTACCGAAGG |
| Sequence-based reagent | Cdh2-RV | Sigma-Aldrich | PCR primers | TCGCTGCTTTCATACTGAACTTT |
| Sequence-based reagent | Coch-FW | Sigma-Aldrich | PCR primers | GTGCAGCAAAACCTGCTACAA |
| Sequence-based reagent | Coch -RV | Sigma-Aldrich | PCR primers | AGCTAGGACGTTCTCTTTGGT |
| Sequence-based reagent | Crabp1-FW | Sigma-Aldrich | PCR primers | CAGCAGCGAGAATTTCGACGA |
| Sequence-based reagent | Crabp1-RV | Sigma-Aldrich | PCR primers | CGCACAGTAGTGGATGTCTTGA |
| Sequence-based reagent | Crp-FW | Sigma-Aldrich | PCR primers | CATAGCCATGGAGAAGCTAC |
| Sequence-based reagent | Crp-RV | Sigma-Aldrich | PCR primers | CAGTGGCTTCTTTGACTCTG |
| Sequence-based reagent | Csn2-FW | Sigma-Aldrich | PCR primers | CTCCACTAAAGGACTTGACAG |
| Sequence-based reagent | Csn2-RV | Sigma-Aldrich | PCR primers | ACCTTCTGAAGTTTCTGCTC |
| Sequence-based reagent | Fabp9-FW | Sigma-Aldrich | PCR primers | CACTGCAGACAACCGAAAAG |
| Sequence-based reagent | Fabp9-RV | Sigma-Aldrich | PCR primers | TCTGTTTGCCAAGCCATTTT |
| Sequence-based reagent | Fam183b-FW | Sigma-Aldrich | PCR primers | CGTGTGGGGCAGATGAAGAAT |

*Continued on next page*

*Continued*

| Reagent type (species) or resource | Designation | Source or reference | Identifiers | Additional information |
|---|---|---|---|---|
| Sequence-based reagent | Fam183b-RV | Sigma-Aldrich | PCR primers | GGTGAATGAGGTTCAGGAACTTG |
| Sequence-based reagent | Fcer2a-FW | Sigma-Aldrich | PCR primers | CCAGGAGGATCTAAGGAACGC |
| Sequence-based reagent | Fcer2a-RV | Sigma-Aldrich | PCR primers | TCGTCTTGGAGTCTGTTCAGG |
| Sequence-based reagent | Fezf2-FW | Sigma-Aldrich | PCR primers | CAGCACTCTCTGCAGACACAA |
| Sequence-based reagent | Fezf2-RV | Sigma-Aldrich | PCR primers | TGCCGCACTGGTTACACTTA |
| Sequence-based reagent | Grap-FW | Sigma-Aldrich | PCR primers | GATCAGGGAGAGTGAGAGTTCC |
| Sequence-based reagent | Grap-RV | Sigma-Aldrich | PCR primers | CAGCTCGTTGAGGGAGTTGA |
| Sequence-based reagent | Icam2-FW | Sigma-Aldrich | PCR primers | ATCAACTGCAGCACCAACTG |
| Sequence-based reagent | Icam2-RV | Sigma-Aldrich | PCR primers | ACTTGAGCTGGAGGCTGGTA |
| Sequence-based reagent | Il15-FW | Sigma-Aldrich | PCR primers | AGCAGATAACCAGCCTACAGGA |
| Sequence-based reagent | Il15-RV | Sigma-Aldrich | PCR primers | TGTTGAAGATGAGCTGGCTATGG |
| Sequence-based reagent | Il21-FW | Sigma-Aldrich | PCR primers | CGCCTCCTGATTAGACTTCG |
| Sequence-based reagent | Il21-RV | Sigma-Aldrich | PCR primers | TGGAGCTGATAGAAGTTCAGGA |
| Sequence-based reagent | Il5-FW | Sigma-Aldrich | PCR primers | CCGCCAAAAAGAGAAGTGTGGCGA |
| Sequence-based reagent | Il5-RV | Sigma-Aldrich | PCR primers | GCCTCAGCCTTCCATTGCCCA |
| Sequence-based reagent | Il7-FW | Sigma-Aldrich | PCR primers | GGGTCCTGGGAGTGATTATGG |
| Sequence-based reagent | Il7-RV | Sigma-Aldrich | PCR primers | CGGGAGGTGGGTGTAGTCAT |
| Sequence-based reagent | Itgad-FW | Sigma-Aldrich | PCR primers | CGAAAGGGTTCAGACTTTGC |
| Sequence-based reagent | Itgad-RV | Sigma-Aldrich | PCR primers | ACACCTCCACGGATAGAAGTC |
| Sequence-based reagent | Itgb6-FW | Sigma-Aldrich | PCR primers | GCTGGTCTGCCTGTTTCTGC |
| Sequence-based reagent | Itgb6-RV | Sigma-Aldrich | PCR primers | TGAGCAGCTTTCTGCACCAC |
| Sequence-based reagent | Kcnj5-FW | Sigma-Aldrich | PCR primers | AAAACCTTAGCGGCTTTGTATCT |
| Sequence-based reagent | Kcnj5-RV | Sigma-Aldrich | PCR primers | AAGGCATTAACAATCGAGCCC |
| Sequence-based reagent | Krt1-FW | Sigma-Aldrich | PCR primers | TGGGAGATTTTCAGGAGGAGG |
| Sequence-based reagent | Krt1-RV | Sigma-Aldrich | PCR primers | GCCACACTCTTGGAGATGCTC |

*Continued on next page*

*Continued*

| Reagent type (species) or resource | Designation | Source or reference | Identifiers | Additional information |
|---|---|---|---|---|
| Sequence-based reagent | Meig1-FW | Sigma-Aldrich | PCR primers | CTTCAGCGGAGGGACAATAC |
| Sequence-based reagent | Meig1-RV | Sigma-Aldrich | PCR primers | CAAGGTTTCAAGGTGGGTGT |
| Sequence-based reagent | Nov-FW | Sigma-Aldrich | PCR primers | AGACCCCAACAACCAGACTG |
| Sequence-based reagent | Nov-RV | Sigma-Aldrich | PCR primers | CGGTAAATGACCCCATCGAAC |
| Sequence-based reagent | Nts-FW | Sigma-Aldrich | PCR primers | GCAAGTCCTCCGTCTTGGAAA |
| Sequence-based reagent | Nts-RV | Sigma-Aldrich | PCR primers | TGCCAACAAGGTCGTCATCAT |
| Sequence-based reagent | Reig1-FW | Sigma-Aldrich | PCR primers | ATGGCTAGGAACGCCTACTTC |
| Sequence-based reagent | Reig1-RV | Sigma-Aldrich | PCR primers | CCCAAGTTAAACGGTCTTCAGT |
| Sequence-based reagent | Resp18-FW | Sigma-Aldrich | PCR primers | CCAGCCAAGATGCAGAGTTCGTTAAAG |
| Sequence-based reagent | Resp18-RV | Sigma-Aldrich | PCR primers | TCAGTCAGCAACAAGGTTGAGGCCCAC |
| Sequence-based reagent | Rsph1-FW | Sigma-Aldrich | PCR primers | ACGGGGACACATATGAAGGA |
| Sequence-based reagent | Rsph1-RV | Sigma-Aldrich | PCR primers | GGCCGTGCTTTTTATTTTTG |
| Sequence-based reagent | Spon2-FW | Sigma-Aldrich | PCR primers | ATGGAAAACGTGAGTCTTGCC |
| Sequence-based reagent | Spon2-RV | Sigma-Aldrich | PCR primers | TGATGCTGTATCTAGCCAGAGG |
| Sequence-based reagent | Sult1c2-FW | Sigma-Aldrich | PCR primers | ATGGCCTTGACCCCAGAAC |
| Sequence-based reagent | Sult1c2-RV | Sigma-Aldrich | PCR primers | TCGAAGGTCTGAATCTGCCTC |
| Sequence-based reagent | Upk3b-FW | Sigma-Aldrich | PCR primers | CATCTGGCTAGTGGTGGCTTT |
| Sequence-based reagent | Upk3b-RV | Sigma-Aldrich | PCR primers | GGTAATGTCATATAGTGGCCGTC |
| Other | Biotinylated Lotus Tetragonolobus Lectin (LTL) | Vector Laboratories | Cat# B-1325; RRID:AB_2336558 | IF (1:500) |
| Other | FITC Ulex Europaeus Agglutinin I (UEA I) | Vector Laboratories | Cat# FL-1061; RRID:AB_2336767 | FACS (1:600) |
| Other | SuperScript II Reverse Transcriptase | Thermo Fisher | Cat# 18064022 | |
| Other | SYBR Premix Ex Taq master mix | Takara | Cat# RR390A | |
| Other | miRNeasy Micro Kit | QIAGEN | Cat# 217084 | |
| Other | TruSeq ChIP Library Preparation Kit | Illumina | Cat# IP-202-2012 | |

## Mice

C57BL/6 WT mice were purchased from Charles River. *Ciita*$^{III+IV-/-}$ (ΔCD4) (*LeibundGut-Landmann et al., 2004*), MHCII$^{-/-}$ (*Madsen et al., 1999*), K14x*Ciita*$^{III\ -/-}$ (mTEC$^{\Delta MHCII}$) (*Irla et al., 2008*), OTII (*Barnden et al., 1998*), RipmOVAxOTII (*Kurts et al., 1996*), and *Rag2*$^{-/-}$ (*Shinkai et al., 1992*) mice were on C57BL/6J background. OTII and RipmOVAxOTII were backcrossed on *Rag2*$^{-/-}$ background. All mice were maintained under specific pathogen-free conditions at an ambient temperature of 22°C at

the animal facilities of the CIML (Marseille, France). Standard food and water were given ad libitum. Males and females were used at the age of 5–6 weeks. All experiments were done in accordance with national and European laws for laboratory animal welfare (EEC Council Directive 2010/63/UE), and were approved by the Marseille Ethical Committee for Animal Experimentation (Comité National de Réflexion Ethique sur l'Expérimentation Animale no. 14).

## mTEC purification

mTECs were isolated by enzymatic digestion with 50 µg/ml of Liberase TM (Roche) and 100 µg/ml of DNase I (Roche) in HBSS medium, as previously described (*Lopes et al., 2017*). CD45$^+$ hematopoietic cells were depleted using anti-CD45 magnetic beads by autoMACS with the depleteS program (Miltenyi Biotec). Total mTECs (EpCAM$^+$UEA-1$^+$Ly51$^{lo}$), mTEC$^{lo}$ (EpCAM$^+$UEA-1$^+$Ly51$^{lo}$CD80$^{lo/int}$), and mTEC$^{hi}$ (EpCAM$^+$UEA-1$^+$Ly51$^{lo}$CD80$^{hi}$) were sorted with a FACSAriaIII cell sorter (BD). The purity of sorted mTEC$^{lo}$ was >98%. Flow cytometry gating strategies are shown in *Figure 1—figure supplement 1*.

## mTEC antigen presentation assays

Variable numbers of mTECs from WT and mTEC$^{\Delta MHCII}$ mice loaded or not with OVA$_{323-339}$ (5 µM, Poly-Peptide group) were co-cultured with 10$^5$ OTII CD4$^+$ T cells (purified with a CD4$^+$ T cell isolation kit, Miltenyi Biotec) in RPMI medium (Thermo Fisher) supplemented with 10% FCS (Sigma-Aldrich), L-glutamine (2 mM, Thermo Fisher), sodium pyruvate (1 mM, Thermo Fisher), 2-mercaptoethanol (2 × 10$^{-5}$ M, Thermo Fisher), penicillin (100 IU/ml, Thermo Fisher), and streptomycin (100 µg/ml, Thermo Fisher). The activation of OTII CD4$^+$ T cells was assessed 18 hr later by flow cytometry based on the upregulation of the CD69 marker.

## Flow cytometry

TECs, thymocytes, and splenic T cells were analyzed by flow cytometry (FACSCanto II, BD) with standard procedures. Cells were incubated for 15 min at 4°C with Fc-block (anti-CD16/CD32, 2.4G2, BD Biosciences) before staining. Antibodies are listed in the Key resources table. For intracellular staining with anti-Foxp3, anti-Ki-67, anti-p38 MAPK, anti-phospho p38 MAPK (Thr180/Tyr182), anti-IKKα, anti-phospho IKKα(Ser180)/IKKβ(Ser181), anti-Erk1/2 MAPK, anti-phospho Erk1/2 MAPK (Thr202/Tyr204), anti-p65, anti-phospho p65(ser536), anti-RelB, anti-phospho RelB(ser552), and anti-DCLK1 antibodies, cells were fixed, permeabilized, and stained with the Foxp3 staining kit according to the manufacturer's instructions (eBioscience). Intracellular staining with anti-Aire, anti-Fezf2, anti-CCL21, anti-H3K4me3, and anti-H3K27me3 antibodies was performed with Fixation/Permeabilization Solution Kit (BD). Secondary antibodies (II Abs) were used to set positive staining gates. Flow cytometry analysis was performed with a FACSCanto II (BD), and data were analyzed using FlowJo software (BD).

## Quantitative RT-PCR

Total RNA was prepared with TRIzol (Invitrogen). cDNAs was synthesized with oligo(dT) using Superscript II reverse transcriptase (Invitrogen). qPCR was performed with the ABI 7500 fast real-time PCR system (Applied Biosystems) and SYBR Premix Ex Taq master mix (Takara). Primers are listed in the Key resources table.

## In vivo transfer of splenocytes into *Rag2*$^{-/-}$ recipients

3.10$^6$ splenocytes purified from the spleen of WT and mTEC$^{\Delta MHCII}$ mice of 8 weeks of age were intravenously injected into *Rag2*$^{-/-}$ female recipients. CD3$^+$, CD4$^+$, and CD8$^+$ T-cell infiltrates were analyzed 6 weeks after transfer by histology and flow cytometry in different peripheral tissues.

## Histology

Tissues were fixed in 10% buffered formalin (Sigma) and embedded in paraffin blocks. 4-µm-thick sections were stained with hematoxylin-eosin (Thermo Fisher) and analyzed by light microscopy (Nikon Statif eclipse Ci-L). For immunofluorescence experiments, frozen thymic sections were stained as previously described (*Sergé et al., 2015*) with Alexa Fluor 488 or Alexa Fluor 647-conjugated anti-Aire (5H12; eBioscience), biotinylated anti-TPA (LTL; Vector Laboratories), rabbit anti-Fezf2 (F441; IBL Tecan), and rabbit anti-involucrin (BioLegend) antibodies. Rabbit anti-Fezf2 and rabbit anti-involucrin

were revealed with Cy3-conjugated anti-rabbit IgG (Invitrogen), and biotinylated anti-TPA was revealed with Alexa Fluor 488-conjugated streptavidin (Invitrogen). Sections were mounted with Mowiol (Calbiochem). Immunofluorescence confocal microscopy was performed with a LSM780 Leica SP5X confocal microscope. Images were analyzed with ImageJ software.

## RNA-seq experiments

Total RNA purified from mTEC$^{lo}$ (*Figure 1—figure supplement 1*) was extracted with miRNeasy Micro Kit (QIAGEN), and RNA quality was assessed on an Agilent 2100 BioAnalyzer (Agilent Technologies). RNA Integrity Number values over 8 were obtained. RNA-seq libraries were generated using the SMART-Seq-v4-Ultra Low Input RNA Kit (Clontech) combined to the Nextera library preparation kit (Illumina) following the manufacturer's instructions. Libraries were sequenced with the Illumina NextSeq 500 machine to generate datasets of single-end 75 bp reads. Two independent biological replicates were used per each condition. RNA-seq data have been deposited with Gene Expression Omnibus (GEO) under the accession number GSE144650.

## RNA-seq analysis

The sequencing reads were mapped to the *Mus musculus* (mm10) reference genome using the TopHat 2 (version 2.0.12) aligner (*Kim et al., 2013*). The reads mapping to the annotated genes (igenome UCSC mm10 GTF: https://support.illumina.com/sequencing/sequencing_software/igenome.html) were counted, normalized, and compared using Cuffdiff2 (version 2.2.1; *Trapnell et al., 2013*) between two conditions. Cuffdiff2 generated expression levels as FPKM, FCs, and p-values to assess the statistical significance of the FPKM difference of each gene between the tested two conditions. Genes showing a significant variation in gene expression between WT and ΔCD4, or WT and mTEC$^{ΔMHCII}$, or RIPmOVAxOTII-*Rag2*$^{-/-}$ and OTII-*Rag2*$^{-/-}$ mice (p-value≤0.05, FC difference ≥ 2 or ≤ 0.5) were considered as up- or downregulated. The TRA and non-TRA gene assignments were obtained from *Sansom et al., 2014*. In this report, the identification of the specificity of expression for each gene in the genome was carried out by analyzing the microarray expression profiles of a large number of different mouse tissues. Aire-dependent, Fezf2-dependent, Aire/Fezf2-dependent, and Aire/Fezf2-independent TRAs were identified using *Aire*$^{-/-}$ mTEC$^{hi}$ RNA-seq datasets and *Fezf2*$^{-/-}$ total mTEC microarray datasets, obtained from the NCBI GEO database (GSE87133 and GSE69105, respectively).

Correlation between the variation of gene expression in mTEC$^{lo}$ from WT versus mTEC$^{ΔMHCII}$ or RIPmOVAxOTII-*Rag2*$^{-/-}$ versus OTII-*Rag2*$^{-/-}$ mice, and of the same genes in mTEC$^{hi}$ from WT versus *Aire*$^{-/-}$ mice was performed doing a locally regression (loess) with the R software (http://www.r-project.org/). Differential gene expression in WT versus *Aire*$^{-/-}$ mTEC$^{hi}$ was obtained by processing using TopHat2 and Cuffdiff2, the sequencing reads corresponding to WT (*Chuprin et al., 2015*) and *Aire*$^{-/-}$ (*Danan-Gotthold et al., 2016*) mTEC$^{hi}$ RNA-seq datasets, which were obtained from the NCBI GEO database (GSE68190 and GSE87133, respectively). HDAC3-dependent mTEC-specific transcription factors regulated by mTEC-thymocyte crosstalk were identified by comparing the top 50 transcriptional regulators that are induced by HDAC3 (*Goldfarb et al., 2016*) with genes upregulated in the different mouse models. TRAs differentially expressed between mTEC$^{lo}$ from WT and mTEC$^{ΔMHCII}$ mice were classified according to their tissue distribution using the mouse ENCODE transcriptome database (*Yue, 2014*). Only tissues that showed the highest expression were taken into account.

## Single-cell RNA-seq analysis

Single-cell RNA-seq count matrix from *Wells et al., 2020* (accession number GSE137699) was reanalyzed with the Seurat package (*Hao et al., 2021*). QC analysis was performed by filtering out cells with a number of feature counts under 200 or over 4000, and a proportion of mitochondrial counts over 4%. Sample integration was performed as described in the Seurat vignette. After PCA for dimension reduction, 15 first dimensions were conserved. Cells were clustered and visualized with UMAP. Cluster annotation was performed by identifying sets of specific markers to each cluster using a differential expression test (FindMarkers function, test = 'roc'). Heatmaps were generated using the pheatmap R package.

## Nano-ChIP-seq experiments

Nano-ChIP-seq was performed as previously described (*Adli and Bernstein, 2011*) on $5.10^4$ purified mTEC$^{lo}$ (*Figure 1—figure supplement 1*). ChIP-seq libraries were prepared with TruSeq ChIP Sample Preparation Kit (Illumina), and 2 × 75 bp paired-end reads were sequenced on an Illumina HiSeq. ChIP-seq data have been deposited with GEO under the accession number GSE144680.

## ChIP-seq analysis

Reads were aligned on the mouse genome (mm10) using Bowtie 2 and default parameters (*Langmead and Salzberg, 2012*). Properly paired alignments were selected using Samtools view with the 0x2flag (-f option). Nonuniquely mapped reads-pairs were filtered out by removing reads with the 'XS' tag set by Bowtie 2. Normalized bedgraphs for ChIP and input samples were generated using MACS2 (*Zhang et al., 2008*) with the callpeak command in BAMPE mode with the --SPMR option. For the diffused H3K27me3 histone mark, the -broad option was used. ChIP enrichment was calculated parsing the ChIP and input normalized bedgraphs with MACS2 and the bdgcmp command (-m FE option). The obtained bedgraphs were converted to wig using the bedGraphToWig.pl script with the --step 10 parameter. MACS2-generated peak calling files were converted to BED files using the cut -f 1-6 command. The obtained Wig and BED files were parsed by CEAS (*Shin et al., 2009*) to generate metagene profile plots corresponding to the average enrichment of H3K4me3 in 3 kb TSS windows or H3K27me3 at gene loci. H3K4me3 and H3K27me3 CEAS-dumped files were parsed to compute ratios of ChIP/input in 1 kb TSS windows and gene loci, respectively. Statistical significance between ChIP enrichment data was tested using the nonparametric Mann–Whitney test. Data were visualized using the Integrative Genomics Viewer (IGV) (*Robinson et al., 2011*).

## Statistics

Data are presented as means ± standard error of mean (SEM). Statistical analysis was performed with GraphPad Prism 7.03 software by using ANOVA, chi-square, unpaired Student's *t*-test, or Mann–Whitney test. ****$p<0.0001$, ***$p<0.001$, **$p<0.01$, *$p<0.05$. Normal distribution of the data was assessed using d'Agostino–Pearson omnibus normality test.

## Acknowledgements

We thank Pr. Arnauld Sergé (LAI, Marseille, France) for critical reading of the manuscript, Pr. Walter Reith (University of Geneva, Switzerland) for providing *Ciita*$^{III+IV-/-}$ and K14x*Ciita*$^{III\ -/-}$ mice, and Dr. Bruno Lucas (Institut Cochin, Paris, France) for providing MHCII$^{-/-}$ mice. We also thank Cloé Zamit and Alexia Borelli (CIML, Marseille, France) for help with mouse genotyping and Lionel Chasson (CIML, Marseille, France) for help with paraffin-embedded tissues. We acknowledge the flow cytometry, the imaging core (ImagImm) and animal facility platforms at CIML for excellent technical support. NL and JC were supported by a PhD fellowship from Aix-Marseille University and the Ministère de l'Enseignement Supérieur et de la Recherche, respectively.

## Additional information

### Funding

| Funder | Grant reference number | Author |
|---|---|---|
| H2020 Marie Skłodowska-Curie Actions | CIG_SIGnEPI4Tol_618541 | Magali Irla |
| Agence Nationale de la Recherche | 2011-CHEX-001-R12004KK | Matthieu Giraud |
| Agence Nationale de la Recherche | ANR-19-CE18-0021-01 RANKLthym | Magali Irla |

The funders had no role in study design, data collection and interpretation, or the decision to submit the work for publication.

## Author contributions
Noella Lopes, Formal analysis, Investigation, Methodology, Validation, Writing – original draft; Nicolas Boucherit, Methodology, Validation; Jérémy C Santamaria, Data curation, Formal analysis, Validation; Nathan Provin, Conceptualization, Data curation, Formal analysis, Visualization; Jonathan Charaix, Matthieu Giraud, Data curation, Formal analysis, Methodology, Software, Validation, Writing - review and editing; Pierre Ferrier, Conceptualization, Funding acquisition, Methodology, Supervision, Writing – original draft, Writing - review and editing; Magali Irla, Conceptualization, Data curation, Formal analysis, Funding acquisition, Methodology, Supervision, Visualization, Writing – original draft, Writing - review and editing

## Author ORCIDs
Noella Lopes http://orcid.org/0000-0002-6296-8426
Jérémy C Santamaria http://orcid.org/0000-0001-7613-3668
Matthieu Giraud http://orcid.org/0000-0002-1208-9677
Magali Irla http://orcid.org/0000-0001-8803-9708

## Ethics
All mice were housed, bred and manipulated under specific pathogen-free conditions at the animal facilities of the CIML (Marseille, France). All experiments were done in accordance with national and European laws for laboratory animal welfare (EEC Council Directive 2010/63/UE), and were approved by the Marseille Ethical Committee for Animal Experimentation (Comité National de Réflexion Ethique sur l'Expérimentation Animale no. 14; Permit Number: 02373.03).

## Decision letter and Author response
Decision letter https://doi.org/10.7554/eLife.69982.sa1
Author response https://doi.org/10.7554/eLife.69982.sa2

# Additional files

## Supplementary files
• Supplementary file 1. List of tissue-restricted self-antigens (TRAs) differentially expressed in mTEC$^{lo}$ from WT and ΔCD4 mice.

• Supplementary file 2. List of tissue-restricted self-antigens (TRAs) differentially expressed in mTEC$^{lo}$ from WT and mTEC$^{\Delta MHCII}$ mice.

• Supplementary file 3. List of tissue-restricted self-antigens (TRAs) differentially expressed in mTEC$^{lo}$ from OTII-$Rag2^{-/-}$ and RipmOVAxOTII-$Rag2^{-/-}$ mice.

• Supplementary file 4. Main target organs and fold change associated with the expression of tissue-restricted self-antigens (TRAs) differentially expressed in mTEC$^{lo}$ from WT and mTEC$^{\Delta MHCII}$ mice.

• Transparent reporting form

## Data availability
RNA-seq data have been deposited in GEO under the accession number GSE144650. ChIP-seq data have been deposited in GEO under the accession number GSE144680.

The following datasets were generated:

| Author(s) | Year | Dataset title | Dataset URL | Database and Identifier |
|---|---|---|---|---|
| Irla M, Giraud M | 2020 | mTEClo/int RNAseq profiling in three mouse models of impaired mTEC/ Tcell crosstalk | https://www.ncbi.nlm.nih.gov/geo/query/acc.cgi?acc=GSE144650 | NCBI Gene Expression Omnibus, GSE144650 |

The following previously published datasets were used:

| Author(s) | Year | Dataset title | Dataset URL | Database and Identifier |
|---|---|---|---|---|
| Wells KL | 2020 | ingle cell sequencing defines a branched progenitor population of stable medullary thymic epithelial cells | https://www.ncbi.nlm.nih.gov/geo/query/acc.cgi?acc=GSE137699 | NCBI Gene Expression Omnibus, GSE137699 |
| Abramson J, Giraud M | 2016 | Aire-KO MEChi RNAseq profiling | https://www.ncbi.nlm.nih.gov/geo/query/acc.cgi?acc=GSE87133 | NCBI Gene Expression Omnibus, GSE87133 |
| Abramson J, Giraud M | 2015 | Sirt1 is essential for Aire-mediated induction of central immunological tolerance | https://www.ncbi.nlm.nih.gov/geo/query/acc.cgi?acc=GSE68190 | NCBI Gene Expression Omnibus, GSE68190 |

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
