## [Editor Report]

This manuscript is of interest to readers in the field of immunology and especially in the induction of immune tolerance in the thymus. The work uses several mouse models to substantially broaden the current understanding of MHCII/TCR-mediated cell-cell crosstalk in the thymus and suggests a novel mechanism that contributes to the generation of functional and self-tolerant T-cells.

---

## [Decision Letter]

**Decision letter after peer review:**

Thank you for submitting your article "Thymocytes trigger self-antigen-controlling pathways in immature medullary thymic epithelial stages" for consideration by *eLife*. Your article has been reviewed by 3 peer reviewers, including Sarah Russell as the Reviewing Editor and Reviewer #1, and the evaluation has been overseen by Tadatsugu Taniguchi as the Senior Editor.

Essential revisions:

With regard to the elucidation of how CD4 cells signal to mTECS to shape their differentiation:

1. According to the recent scRNA sequencing studies (reviewed in Kadouri et al. 2020), the mTEClow mTECs contain at least two distinct subpopulations: the functionally mature CCL21-producing mTEC I and the immature mTECs giving rise to mTEC II and III. In its current form, however, the manuscript largely ignores the presence of mTEC I cells. It would be important to analyze the changes in this population in the knockout models (by sequencing and/or qPCR) and cover this population also in the introduction and discussion.

2. The authors conclude that their RNAseq data in figures 1 and 4 show that genes are upregulated/downregulated. However, it could also be that their differential gene/cytokine expression is due to the presence of different mTEC^lo^ subsets, and the authors show this in figure 3. This would change the conclusion to: CD4^+^ thymocytes alter mTEC^lo^ differentiation states, associated with differential gene expression. This is also the case in figure 2. For instance, Lopez et al. state that AIRE expression 4.5-fold higher in mTEC^ΔMHCII^ cells but then they show that there are different percentages of AIRE^+^ cells (change in the mTEC^lo^ subsets in the ko mice). It is important to make clear that the signalling/expression data are confounded by differences in mTEC cell populations changes might reflect different cell composition rather than different signalling or expression states.

3. In figure 3, the authors show mTEC^hi^ cells in dCD4 and mTEC^ΔMHCII^ mice. It is surprising that mTEC^hi^ cells are present since these shouldn't differentiate in the absence of an MHCII-TCR interaction. It is important to address how these cells develop and whether their presence alters interpretation of the results.

With regard to how these interactions shape the TCRVβ repertoire in mature T cells:

4. Differences in the percentage of a few TCRVβ families are not sufficient to conclude that there is an alteration in the TCRVβ repertoire. Sequencing the different TCRs and evaluate constraints on TCR-CDR3 segments will be required for this conclusion.

*Reviewer #1 (Recommendations for the authors):*

The assumption by the authors that the observed phenotypes are driven by CD4 deficiency is central to the conclusions of the manuscript, but is not tested in the manuscript. This seems an important caveat to address.

The authors should make clear whether the genetic disruptions they use directly or indirectly influence the cell types that they claim to be relevant for each phenotype, and what other alternative explanations might lead to the phenotypes.

*Reviewer #2 (Recommendations for the authors):*

Occasionally (as in page 12 para 2) the authors interpret a decrease (of transcription factors) in their knockout model as an increase in wild-type mouse. Although hypothetically right, this should be corrected especially in the Results section.

*Reviewer #3 (Recommendations for the authors):*

1. The authors present data demonstrating that the lack of CD4 on thymocytes changes NFkb signalling in mTEC^lo^ cells, with decreased IKKa/p38 signalling and increased pRELB. MFI would likely be better evaluated as a staining index or ratio, to control staining in that mouse type.

2. Staining for DclK^+^ cells in the AIRE- subset should show all gates.

3. In figure 4, the staining for pRELB relative to total RELB would suggest higher phosphorylation in the OTII-RAG2^-/-^ mice (i.e., total RELB is more decreased than pRELB). Staining for RelB appears very heterogeneous, suggesting that it is expressed at different levels in different subsets. Also, staining controls for both types of mice are not shown. This is also the case in figure 6C.

---

## [Author Response]

Essential revisions:With regard to the elucidation of how CD4 cells signal to mTECS to shape their differentiation:1. According to the recent scRNA sequencing studies (reviewed in Kadouri et al. 2020), the mTEClow mTECs contain at least two distinct subpopulations: the functionally mature CCL21-producing mTEC I and the immature mTECs giving rise to mTEC II and III. In its current form, however, the manuscript largely ignores the presence of mTEC I cells. It would be important to analyze the changes in this population in the knockout models (by sequencing and/or qPCR) and cover this population also in the introduction and discussion.

We thank the Reviewers for highlighting that mTEC^lo^ also contain CCL21-producing cells, called mTEC I. In this revised version, we analyzed CCL21^+^ mTEC^lo^ by flow cytometry in the different mouse models described in this study, i.e., ΔCD4 (absence of CD4^+^ thymocytes), mTEC^ΔMHCII^ (absence of MHCII-TCR interactions between mTEC and CD4^+^ thymocytes), MHCII^-/-^, OTII and RipmOVA x OTII mice. As shown in the new Figure 3D and Figure 3—figure supplement 3B, the frequencies of CCL21^+^ cells among mTEC^lo^ cells were globally unaffected in ΔCD4, mTEC^ΔMHCII^ and MHCII^-/-^ mice, although their numbers were reduced, consistently with a smaller medulla compartment in these mice, previously reported by our group (cf. Irla et al. Plos One 2012 and J Immunol. 2013). Numbers of CCL21^+^ mTEC^lo^ were also increased in RipmOVA x OTII mice compared to OTII mice (cf. New Figure 5G), consistently with our previous observation that antigen-specific interactions between mTECs and CD4^+^ thymocytes induce medulla development (cf. Irla et al. Plos One 2012).

We now cover the CCL21^+^ mTEC^lo^ subset in the introduction and discuss these new results.

2. The authors conclude that their RNAseq data in figures 1 and 4 show that genes are upregulated/downregulated. However, it could also be that their differential gene/cytokine expression is due to the presence of different mTEC^lo^ subsets, and the authors show this in figure 3. This would change the conclusion to: CD4^+^ thymocytes alter mTEC^lo^ differentiation states, associated with differential gene expression. This is also the case in figure 2. For instance, Lopez et al. state that AIRE expression 4.5-fold higher in mTEC^ΔMHCII^ cells but then they show that there are different percentages of AIRE^+^ cells (change in the mTEC^mhcii^subsets in the ko mice). It is important to make clear that the signalling/expression data are confounded by differences in mTEC cell populations changes might reflect different cell composition rather than different signalling or expression states.

We fully agree that mTEC^lo^ cells are heterogeneous and that changes in mTEC^lo^ subset composition in the transgenic mouse models analyzed could contribute to the differences observed. To clarify this point, we aimed at investigating in this revised version whether the upregulation/downregulation of genes observed by RNAseq in figures 1, 2 and 4 could be due to the impact of the crosstalk at the gene expression level or rather due to the presence of different mTEC^lo^ subsets. To this end, we analyzed single cell RNA-seq data on mTEC recently published by Wells KL et al. (*eLife*. 2020) (cf. new Figure 1—figure supplement 4) to determine whether the genes found to be affected by the thymic crosstalk, are specific to a particular mTEC subset (i.e. CCL21^+^, TAC-TEC, Aire^+^, Post-Aire and Tuft-like cells). In accordance with the reduced proportions of Post-Aire and DCLK1^+^ Tuft-like cells observed respectively by histology and flow cytometry (cf. Figure 3E, Figure 5H, Figure 3—figure supplement 2A and Figure 5—figure supplement 3A), some of these genes were associated with Post-Aire and Tuft-like cells (cf. new Figure 1I, new Figure 2K, new Figure 4L). Importantly, this new analysis revealed that many genes upregulated by CD4^+^ thymocytes were highly expressed by Aire^+^ mTECs, strongly suggesting that their increase in mTEC^lo^ is due to an enhanced transcriptional activity accompanying the transition to Aire^+^ mTECs rather than changes in mTEC^lo^ subset composition. In line with this hypothesis, we found that several of these genes were already expressed in TAC-TECs that precede Aire^+^ mTECs.

These new results, suggesting a direct upregulation of a number of mTEC^hi^-specific genes in mTEC^lo^, as well as the possibility that signaling data could be confounded by differences in mTEC^lo^ subsets, are now discussed in this revised version.

3. In figure 3, the authors show mTEC^hi^ cells in dCD4 and mTEC^ΔMHCII^ mice. It is surprising that mTEC^hi^ cells are present since these shouldn't differentiate in the absence of an MHCII-TCR interaction. It is important to address how these cells develop and whether their presence alters interpretation of the results.

To clarify this point, we now show frequencies and numbers of mTEC^hi^ analyzed by flow cytometry in WT, dCD4 and mTEC^ΔMHCII^ mice (cf. new Figure 3A). The development of these cells is substantially impaired but not totally abrogated in these mice, which is in total agreement with previous observations from our group (Irla et al. Immunity 2008 and Plos One 2012). Since the development of mTEC^hi^, including Aire^+^ cells, relies on RANK signaling (Khan et al. J Exp Med. 2014, Akiyama et al. Immunity 2008, Hikosaka et al. Immunity 2008, Rossi et al. J Exp Med. 2007, Irla Front Immunol 2021), the residual development of mTEC in dCD4 and mTEC^ΔMHCII^ mice could be explained by the fact that in addition to CD4^+^ thymocytes, RANKL is also expressed by invariant NKT cells, which are present in these mice. In agreement with this hypothesis, invariant NKT cells have been shown to participate in Aire^+^ mTEC differentiation in the post-natal thymus, but to a lesser extent than CD4^+^ thymocytes (White AJ et al. J Immunol 2014). We clarified this point in the discussion (lines 441-444).

The residual development of mTEC^hi^ in these mice does not interfere with the interpretation of the results. Indeed, we took into consideration the impact of the residual development of mTEC^hi^ in the composition of mTEC^lo^ by analyzing post-Aire cells that derived from Aire^+^ mTEC^hi^ (cf. Figure 3—figure supplement 2A and Figure 5—figure supplement 3A). In agreement with the impaired development of Aire^+^ mTEC^hi^ in dCD4, mTEC^ΔMHCII^ mice, numbers of Involucrin^+^TPA^+^Aire^-^ post-Aire cells were reduced. In this revised version, we also analyzed whether genes found to be differentially expressed by RNA-seq are associated with post-Aire cells using single-cell RNAseq data (cf. new Figures 1I, 2K and 4L). In line with the reduction in post-Aire cells, some differentially expressed genes were associated with these cells.

With regard to how these interactions shape the TCRVβ repertoire in mature T cells:4. Differences in the percentage of a few TCRVβ families are not sufficient to conclude that there is an alteration in the TCRVβ repertoire. Sequencing the different TCRs and evaluate constraints on TCR-CDR3 segments will be required for this conclusion.

We fully agree that the differences observed by flow cytometry in the percentages of TCRVβ usage are insufficient to conclude that the TCRVβ repertoire is altered. We have modified the text in the Results section (cf. lines 97, 368) and discussed that future experiments based on TCR sequencing are expected to clarify this point (cf. lines 482-489), which is beyond the scope of the current study.

Reviewer #1 (Recommendations for the authors):The assumption by the authors that the observed phenotypes are driven by CD4 deficiency is central to the conclusions of the manuscript, but is not tested in the manuscript. This seems an important caveat to address.The authors should make clear whether the genetic disruptions they use directly or indirectly influence the cell types that they claim to be relevant for each phenotype, and what other alternative explanations might lead to the phenotypes.

We understand the point raised by the Reviewer that a genetic mutation in the class II transactivator CIITA in dCD4 and mTEC^ΔMHCII^ mice could indirectly influence the mTEC subsets analyzed. This is the reason why we have completed our study by analyzing mTEC^lo^ from MHCII^-/-^ mice both by qPCR and flow cytometry (cf. Figure 1—figure supplement 3A and Figure 3—figure supplement 3). Similar defects in TRA expression and in the cellularity of mTEC subsets (i.e. Aire^-^Fezf2^+^ and Aire^+^Fezf2^+^ mTECs, as well as in CCL21^+^ and DCLK1^+^ mTEC^lo^) were observed in these mice, ruling out any potential indirect effect of CIITA in the observed results (cf. lines 132-134).

Furthermore, to strengthen our general conclusion that self-reactive CD4^+^ thymocytes have a direct effect on mTEC and further exclude any potential indirect impact of CIITA or even MHCII molecules in the observed results, we compared the transcriptome of mTEC^lo^ from OTII-Rag2^-/-^ and RipmOVA x OTII-Rag2^-/-^ mice by RNA-seq and analyzed their respective composition in mTEC subsets by flow cytometry. Importantly, the only difference in these two transgenic mice relies on the expression (RipmOVA x OTII-Rag2^-/-^ mice) or not (OTII-Rag2^-/-^ mice) of the cognate OVA antigen, which is driven by the rat insulin promoter (Rip) specifically in mTEC. Again, the analysis of these mice led to the same overall phenotype as observed in the WT vs. dCD4 or the WT vs. mTEC^ΔMHCII^ comparison.

In summary, the comparison of these three distinct transgenic models by the different approaches used strongly supports the notion that self-reactive CD4^+^ thymocytes directly affect the mTEC subsets analyzed.

Reviewer #2 (Recommendations for the authors):Occasionally (as in page 12 para 2) the authors interpret a decrease (of transcription factors) in their knockout model as an increase in wild-type mouse. Although hypothetically right, this should be corrected especially in the Results section.

We accordingly modified the text (cf. Lines 189-194).

Reviewer #3 (Recommendations for the authors):1. The authors present data demonstrating that the lack of CD4 on thymocytes changes NFkb signalling in mTEC^lo^ cells, with decreased IKKa/p38 signalling and increased pRELB. MFI would likely be better evaluated as a staining index or ratio, to control staining in that mouse type.

Thank you for this comment. We now represent MFI as a ratio to secondary antibody control staining (cf. new Figures 1A and 4B).

2. Staining for DclK^+^ cells in the AIRE- subset should show all gates.

We now show the gating strategy concerning the identification of DCLK1^+^ mTEC (cf. new Figure 3—figure supplement 2B and new Figure 5—figure supplement 3B).

3. In figure 4, the staining for pRELB relative to total RELB would suggest higher phosphorylation in the OTII-RAG2^-/-^ mice (i.e., total RELB is more decreased than pRELB). Staining for RelB appears very heterogeneous, suggesting that it is expressed at different levels in different subsets. Also, staining controls for both types of mice are not shown. This is also the case in figure 6C.

To clarify whether the heterogeneous staining of RelB expression observed by flow cytometry could be due to different expression levels in distinct mTEC^lo^ subsets, we analyzed *Relb* gene expression level in CCL21^+^, TAC-TEC, post-Aire and Tuft-like mTEC using single-cell RNA-seq data. We found that *Relb* is highly expressed in TAC-TEC and post-Aire cells (cf. New Figure 4-supplement 2A). These observations are in agreement with the hypothesis that self-reactive CD4^+^ thymocytes activate NFkB signaling from the TAC-TEC stage that precedes Aire^+^ mTECs and that this activation persists in post-Aire cells. We modified the text accordingly (cf. lines 259-263).

Unfortunately, it is technically not possible to detect RelB in Tuft-like mTEC because anti-RelB and anti-DCLK1 antibodies are both produced in rabbit and are detected through anti-rabbit secondary antibodies. It is also not possible to evaluate RelB level in post-Aire and TAC-TEC cells because there is no valuable markers to date that allow the identification of these cells by flow cytometry. Therefore, we discuss that we cannot exclude that the heterogeneity in RelB expression could also be due to different representation of mTEC^lo^ subsets (cf. lines 396-400).

Moreover, we verified that the secondary antibody control staining gives the same signal in RipmOVA x OTII-Rag2^-/-^ and OTII-Rag2^-/-^ mice (Author response image 1).

**Author response image 1. sa2fig1:**